# A²Q: Aggregation-Aware Quantization for Graph Neural Networks

**Zeyu Zhu**[1,2]    **Fanrong Li**[1]    **Zitao Mo**[1]    **Qinghao Hu**[1]    **Gang Li**[4]    **Zejian Liu**[1,2]
**Xiaoyao Liang**[4]    **Jian Cheng**[1,2,3*]
[1]Institute of Automation, Chinese Academy of Sciences
[2]School of Future Technology, University of Chinese Academy of Sciences
[3]AiRiA
[4]Shanghai Jiao Tong University
{zhuzeyu2021, lifanrong2017, mozitao2017}@ia.ac.cn,
{huqinghao2014, liuzejian2018}@ia.ac.cn, {gliaca}@sjtu.edu.cn,
{liang-xy}@cs.sjtu.edu.cn, {jcheng}@nlpr.ia.ac.cn

## Abstract

As graph data size increases, the vast latency and memory consumption during inference pose a significant challenge to the real-world deployment of Graph Neural Networks (GNNs). While quantization is a powerful approach to reducing GNNs complexity, most previous works on GNNs quantization fail to exploit the unique characteristics of GNNs, suffering from severe accuracy degradation. Through an in-depth analysis of the topology of GNNs, we observe that the topology of the graph leads to significant differences between nodes, and most of the nodes in a graph appear to have a small aggregation value. Motivated by this, in this paper, we propose the Aggregation-Aware mixed-precision Quantization (A²Q) for GNNs, where an appropriate bitwidth is automatically learned and assigned to each node in the graph. To mitigate the vanishing gradient problem caused by sparse connections between nodes, we propose a Local Gradient method to serve the quantization error of the node features as the supervision during training. We also develop a Nearest Neighbor Strategy to deal with the generalization on unseen graphs. Extensive experiments on eight public node-level and graph-level datasets demonstrate the generality and robustness of our proposed method. Compared to the FP32 models, our method can achieve up to a 18.6x (i.e., 1.70bit) compression ratio with negligible accuracy degradation. Morever, compared to the state-of-the-art quantization method, our method can achieve up to 11.4% and 9.5% accuracy improvements on the node-level and graph-level tasks, respectively, and up to 2x speedup on a dedicated hardware accelerator.

## 1 Introduction

Recently, Graph Neural Networks (GNNs) have attracted much attention due to their superior learning and representing ability for non-Euclidean geometric data. A number of GNNs have been widely used in real-world applications, such as recommendation system (Jin et al., 2020), and social network analysis (Lerer et al., 2019), etc. Many of these tasks put forward high requirements for low-latency inference. However, the real-world graphs are often extremely large and irregular, such as Reddit with 232,965 nodes, which needs 19G floating-point operations (FLOPs) to be processed by a 2-layer Graph Convolutional Network (GCN) with only 81KB parameters (Tailor et al., 2020), while ResNet-50, a 50-layer DNN, only takes 8G FLOPs to process an image (Canziani et al., 2016). What is worse, it requires a huge amount of memory access for GNNs inference, e.g., the nodes features size of Reddit is up to 534MB, leading to high latency. Therefore, the aforementioned problems pose a challenge to realize efficient inference of GNNs.

Neural network quantization can reduce the model size and accelerate inference without modifying the model architecture, which has become a promising method to solve this problem in recent years.

---

*Corresponding author

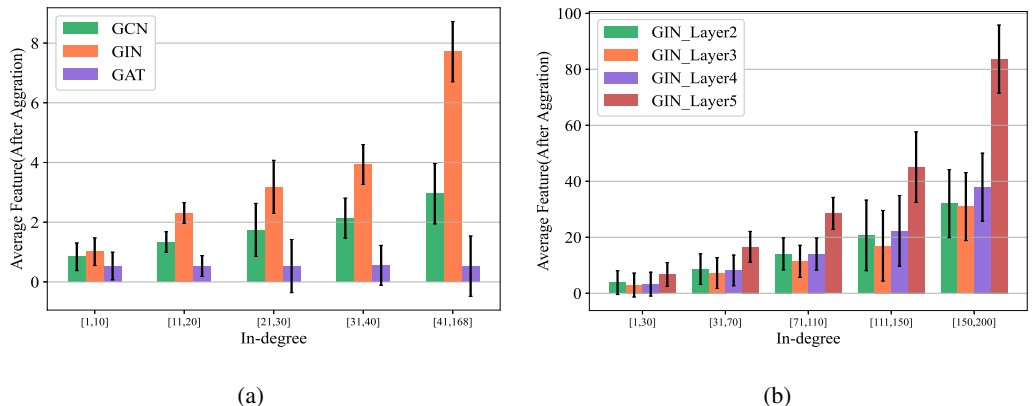

Figure 1: The analysis of the average aggerated node features in different in-degrees node groups on various tasks. (a) The values at the final layer for GNNs trained on Cora. (b) The values at the 2-5 layer of GIN trained on REDDIT-BINARY. The average values are all generated from 10 runs.

Unfortunately, there remain some issues in the existing works on GNNs quantization. Feng et al. (2020) only quantizes the node feature and keeps floating point calculations during inference. Tailor et al. (2020) proposes a degree-quant training strategy to quantize GNNs to the low-bit fixed point but causes a large accuracy drop, e.g., 11.1% accuracy drops when quantizing to 4bits. Moreover, some works (Wang et al., 2021b; Bahri et al., 2021; Wang et al., 2021a; Jing et al., 2021) quantize GNNs into 1-bit and compute with XNOR and bit count operations. However, these 1-bit quantization methods are either restricted to the node-level tasks or can not generalize well to other GNNs.

Most of the above methods do not make full use of the property of GNNs and graph data, resulting in severe accuracy degradation or poor generalization. As presented in MPNN framework (Gilmer et al., 2017), GNNs processing is divided into two phase: First, in the aggregation phase, a node collects information from neighboring nodes and uses the aggregation function to generate hidden features; second, in the update phase, the hidden features are transformed into new features by an update function. We analyze the nodes features after aggregation in Figure 1 and find that the higher the in-degree is, the larger the node features tend to be after aggregation. And the features vary significantly between nodes with different in-degrees, which represent the topology of a graph. Moreover, according to Xie et al. (2014); Aiello et al. (2001), the degrees of nodes in most real-world graph data often follow the power-law distribution, i.e., nodes with a low degree account for the majority of graph data. Therefore, specially quantizing the nodes features according to the topology of the graphs will be beneficial to reduce the quantization error while achieving a higher compression ratio.

In this paper, we propose the **Aggregation-Aware Quantization** ($A^2Q$) method, which quantizes different nodes features with different learnable quantization parameters, including bitwidth and step size. These parameters can be adaptively learned during training and are constrained by a penalty on memory size to improve the compression ratio. However, when quantizing the model in semi-supervised tasks, the gradients for most quantization parameters are zero due to the sparse connections between nodes, which makes the training non-trivial. We propose the **Local Gradient** method to solve this problem by introducing quantization error as supervised information. Finally, to generalize our method to unseen graphs in which the number of the nodes varies, we develop the **Nearest Neighbor Strategy** which assigns the learned quantization parameters to the unseen graph nodes. To the best of our knowledge, we are the first to introduce the mixed-precision quantization to the GNNs. Compared with the previous works, our proposed methods can significantly compress GNNs with negligible accuracy drop.

In summary, the key contributions of this paper are as follows:

1) We propose the Aggregation-Aware mixed-precision Quantization ($A^2Q$) method to enable an adaptive learning of quantization parameters. Our learning method is powerful by fully utilizing the characteristic of GNNs, and the learned bitwidth is strongly related to the topology of the graph.

2) A **Local Gradient** method is proposed to train the quantization parameters in semi-supervised learning tasks. Furthermore, to generalize our method to the unseen graphs in which the number of input nodes is variable, we develop the **Nearest Neighbor Strategy** to select quantization parameters for the nodes of the unseen graphs.

3) Experiments demonstrate that we can achieve a compression ratio up to 18.6x with negligible accuracy degradation compared to the full-precision (FP32) models. Moreover, the model trained with our $A^2Q$ method outperforms the state-of-the-art (SOTA) method up to 11.4% with a speedup up to 2.00x in semi-supervised tasks, and obtains up to 9.5% gains with a 1.16x speedup in graph-level tasks. We provide our code at this URL: https://github.com/weihai-98/$A^2$Q.

## 2 RELATED WORK

***Graph Neural Networks:*** The concept of the graph neural network was first proposed in Scarselli et al. (2008), which attempted to generalize neural networks to model non-Euclidean data. In the following years, various GNN models were proposed. For example, Graph Convolution Network (GCN) (Kipf & Welling, 2016) uses a layer-wise propagation rule that is based on a first-order approximation of spectral convolutions on graphs, Graph Isomorphism Network (GIN) (Xu et al., 2018) designed a provably maximally powerful GNN under the MPNN framework, and Graph Attention Network (GAT) (Veličković et al., 2017) introduces the attention mechanism to graph processing. Although GNNs have encouraging performance in a wide range of domains (Jin et al., 2020; Yang, 2019), the huge amount of float-point operations and memory access in process pose a challenge to efficient inference, which hinder the applications of GNNs.

***Quantized GNNs:*** As a promising method to reduce the model size and accelerate the inference process, quantization is also applied to GNNs. Some works quantize features and weights in GNNs to low bitwidths (Feng et al., 2020; Tailor et al., 2020) or even 1-bit (Wang et al., 2021b; Bahri et al., 2021; Wang et al., 2021a; Jing et al., 2021), i.e., use fixed-point numbers instead of floating-point numbers for computation. But when the compression ratio is high (e.g., <4bit), the performance degradation of these works is significant, and the generalization of 1-bit method is limited. There are also some works on vector quantization (VQ), which use the vectors in a codebook obtained during the training process instead of the original features (Ding et al., 2021; Huang et al., 2022). However, searching for vectors in the codebook is computationally complex.

***Mixed-Precision Quantization:*** Based on the idea that different layers have different sensitivities to quantization, mixed-precision quantization is proposed in CNNs to quantize different layers to different bitwidths for better model compression. Early works (Wang et al., 2019; Lou et al., 2019) proposed reinforcement learning (RL) based methods to search bitwidth for different layers, but they often require large computational resources, which limits the exploration of the search space. Another important class of mixed-precision method is the criteria-based method, they use the specific criteria to represent the quantization sensitivity, e.g., (Dong et al., 2019; 2020; Chen et al., 2021)quantize different layers with different bitwidths based on the trace of the Hessian. Recently, there are some other methods to learn the bitwidth during training (Uhlich et al., 2019; Esser et al., 2019; Jain et al., 2020). However, due to the huge difference between GNNs and CNNs, it is difficult to use these methods on GNNs directly, and our $A^2Q$ is the first method to introduce the mixed-precision quantization to GNNs, further improving the inference efficiency of GNNs.

## 3 METHOD

In this section, we describe our proposed Aggregation-Aware Quantization in detail. Firstly, we present the formulation of the mixed-precision quantization for GNNs, which fully utilizes the property of GNNs and graph data. Secondly, we introduce the Local Gradient method to address the gradient vanishing problem during training. Finally, we detail the Nearest Neighbor Strategy, which is used for generalizing our approach to the unseen graphs.

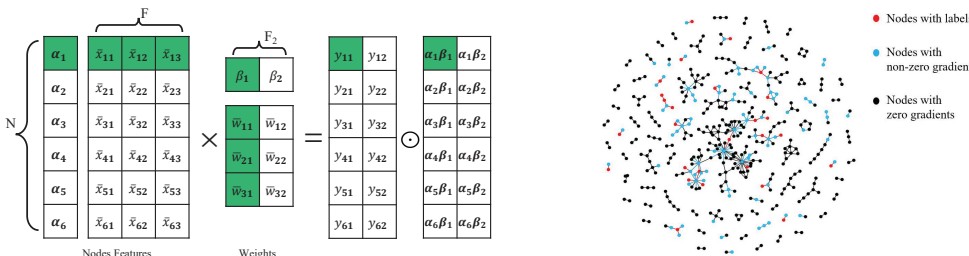

Figure 2: Perform matrix multiplication by the integer represented. $\bar{x}$ and $\bar{w}$ are both integers.

Figure 3: The gradients to $x_q$ in GCN trained on Cora by sampling 400 nodes.

## 3.1 AGGREGATION-AWARE QUANTIZATION

We assume a graph data with $N$ nodes and the node features are $F$-dimensional, i.e., the feature map is $\boldsymbol{X} \in \mathbb{R}^{N \times F}$ and $\boldsymbol{x}_i$ is the features of node $i$. We use the learnable parameters step size $\alpha_i \in \mathbb{R}_+$ and bitwidth $b_i \in \mathbb{R}_+$ to quantize the features of the $i$-th node as:

$$\bar{\boldsymbol{x}}_i = sign(\boldsymbol{x}_i) \begin{cases} \lfloor \frac{|\boldsymbol{x}_i|}{\alpha_i} + 0.5 \rfloor, & |\boldsymbol{x}| < \alpha_i(2^{[b_i]-1} - 1) \\ \\ 2^{[b_i]-1} - 1, & |\boldsymbol{x}_i| \geq \alpha_i(2^{[b_i]-1} - 1) \end{cases}, \tag{1}$$

where $\lfloor \cdot \rfloor$ is the floor function, and $[\cdot]$ is the round function to ensure the bitwidth used to quantize is an integer. The learnable parameters are $\boldsymbol{s}_X = (\alpha_1, \alpha_2, ..., \alpha_N)$, and $\boldsymbol{b}_X = (b_1, b_2, ..., b_N)$. Then we can obtain the fixed-point feature map $\bar{\boldsymbol{X}}$, and the original feature can be represented as $\boldsymbol{X}_q = \boldsymbol{S}_X \cdot \bar{\boldsymbol{X}}$, where $\boldsymbol{S}_X = diag(\alpha_1, \alpha_2, ..., \alpha_N)$. Note that we use $[b] + 1$ as the quantization bitwidth for the features after ReLU because the values are all non-negative.

In the update phase, the node features are often transformed with a linear mapping or an MLP in which matrix multiplication $\boldsymbol{XW}$ is the main computation, and the transformed node features are the input to the next layer in GNNs. In order to accelerate the update phase, we also quantize $\boldsymbol{W}$. Due to the fact that $\boldsymbol{W}$ in a certain layer is shared by all nodes, we quantize $\boldsymbol{W}$ to the same bitwidth of 4bits for all GNNs in this paper. However, each column of $\boldsymbol{W}$ has its learnable quantization step size, i.e., $\boldsymbol{s}_W = (\beta_1, \beta_2, .., \beta_{F_2})$, where $F_2$ is the output-dimension of the node features in current layer and $\beta_i$ is the quantization step size for the $i$-th column of $\boldsymbol{W}$, and we also use Eq. 1 to quantize $\boldsymbol{W}$. We can obtain the integer representation $\bar{\boldsymbol{W}}$ and the quantized representation $\boldsymbol{W}_q = \bar{\boldsymbol{W}} \cdot \boldsymbol{S}_W$, where $\boldsymbol{S}_W = diag(\beta_1, \beta_2, ..., \beta_{F_2})$. The float-point matrix multiplication in the update phase can be reformulated as follow:

$$\boldsymbol{X} \cdot \boldsymbol{W} \approx \boldsymbol{X}_q \cdot \boldsymbol{W}_q = (\boldsymbol{S}_X \cdot \bar{\boldsymbol{X}}) \cdot (\bar{\boldsymbol{W}} \cdot \boldsymbol{S}_W) = (\bar{\boldsymbol{X}} \cdot \bar{\boldsymbol{W}}) \odot (\boldsymbol{s}_X \otimes \boldsymbol{s}_W), \tag{2}$$

where $\odot$ denotes an element-wise multiplication, and $\otimes$ denotes the outer product. After training, we can obtain $\boldsymbol{s}_X$ and $\boldsymbol{s}_W$ so that the outer product can be pre-processed before inference. An example is illustrated in Figure 2. For the aggregation phase, i.e., $\boldsymbol{AX}$, $\boldsymbol{A}$ is the adjacency matrix and $\boldsymbol{A} \in \{0,1\}^{N \times N}$, we quantize the $\boldsymbol{X}$ as the quantization way of $\boldsymbol{W}$ because the nodes features involved in the aggregation process come from the update phase, in which the features lose the topology information of graphs. Then the aggregation phase can be performed by integer operations to reduce the computational overhead.

The quantization parameters $(s, b)$ are trained by the backpropagation algorithm. Since the floor and round functions used in the quantization process are not differentiable, we use the straight-through estimator (Bengio et al., 2013) to approximate the gradient through these functions, and the gradients of the quantization parameters can be calculated by:

$$\frac{\partial L}{\partial s} = \sum_{i=1}^{d} \frac{\partial L}{\partial x_q^i} \cdot \frac{\partial x_q^i}{\partial s}, \tag{3} \qquad\qquad \frac{\partial L}{\partial b} = \sum_{i=1}^{d} \frac{\partial L}{\partial x_q^i} \cdot \frac{\partial x_q^i}{\partial b}, \tag{4}$$

where $d$ is the dimension of the vector $\boldsymbol{x}$, $(s, b)$ are the quantization parameters for $\boldsymbol{x}$, and $x_q^i$ is the value of $i$-th dimension in $\boldsymbol{x}_q$. Detailed information about quantization process and the backpropagation are shown in Appendix A.1 and A.3 **Proof 2 and 3**.

In order to improve the compression ratio of the node features, we introduce a penalty term on the memory size:

$$L_{memory} = (\frac{1}{\eta} \cdot \sum_{l=1}^{L} \sum_{i=1}^{N} \dim^l \cdot b_i^l - M_{target})^2 \ , \tag{5}$$

where $L$ is the number of layers in the GNNs, $N$ is the total number of nodes, $\dim^l$ is the length of the node features in $l$-th layer, $b_i^l$ is the quantization bitwidth for node $i$ in $l$-th layer, $M_{target}$ is the target memory size on the total node features memory size, and $\eta = 8 * 1024$, which is a constant to convert the unit of memory size to KB. Then the model and quantization parameters can be trained by the loss function:

$$L_{total} = L_{task} + \lambda \cdot L_{memory} \ , \tag{6}$$

where $L_{task}$ is the task-related loss function and $\lambda$ is a penalty factor on $L_{memory}$.

## 3.2 LOCAL GRADIENT

Although the above end-to-end learning method is concise and straightforward, the gradients for the quantization parameters of nodes features, i.e., $\frac{\partial L_{task}}{\partial s}$ and $\frac{\partial L_{task}}{\partial b}$, are almost zero during the training process of semi-supervised tasks, which poses a significant challenge to train the quantization parameters for nodes features. We analyze the property of GNNs and graph data, and find that two reasons lead to this phenomenon: 1. The extreme sparsity of the connections between nodes in graph data. 2. Only a tiny fraction of nodes with labels are used for training in semi-supervised tasks (e.g., 0.30% in PubMed dataset). Therefore, $\frac{\partial L_{task}}{\partial x_q}$ for most node features are zero (detailed proof in Appendix A.3.2), which results in that the gradients for quantization parameters of these nodes vanish according to Eq. 3 and Eq. 4. To clarify, we visualize the $\frac{\partial L_{task}}{\partial x_q}$ in the second layer of GCN trained on Cora. As shown in Figure 3, most gradients for the nodes features are zero.

The gradients of the $L_{task}$ w.r.t. quantized nodes features can be viewed as the supervised information from the labeled nodes which enable the training of the quantization parameters for nodes features. However, this supervised information is missing due to zero gradients. Considering the quantization error is related to the $L_{task}$, we introduce the quantization error $E = \frac{1}{d} |x_q - x|_1$ as the supervised information for the quantization parameters of nodes features, where $x$ is the features before quantization, $x_q$ is the features after quantization and $|\cdot|_1$ denotes the L1 norm. We refer to this method as **Local Gradient** because the gradients are computed by the local quantization errors instead of back-propagated task-related gradients. Then the quantization parameters for node features can be trained by gradients from $E$:

$$\frac{\partial E}{\partial s} = \frac{1}{d} \sum_{i=1}^{d} sign(x_q^i - x^i) \cdot \frac{\partial x_q^i}{\partial s} \ , \quad (7) \qquad \frac{\partial E}{\partial b} = \frac{1}{d} \sum_{i=1}^{d} sign(x_q^i - x^i) \cdot \frac{\partial x_q^i}{\partial b} \ . \quad (8)$$

Note that the quantization parameters of $W$ are still trained by utilizing the gradients in Eq. 3.

## 3.3 NEAREST NEIGHBOR STRATEGY

In graph-level tasks, the quantized GNNs are required to generalize to unseen graphs. In such a scenario, the number of input nodes may vary during training or inference. However, the learnable method can only train a fixed number of $(s, b)$ pairs which are the same as the number of input nodes, so it is challenging to learn the $s$ and $b$ for every node in graph-level tasks. To solve this problem, we propose the **Nearest Neighbor Strategy**, which allows learning of a fixed number of quantization parameters and select quantization parameters for the unseen graphs.

The proposed strategy is shown in Algorithm 1. To ensure the numerical range of $x_q$ is as close as to $x$ at FP32, a simple way is to keep the maximum quantization value equal to the maximum absolute value of $x$. Based on this idea, we first initialize $m$ groups of quantization parameters, then we calculate the maximum quantization value for every group, i.e., $q_{max} = s(2^{[b]-1} - 1)$. When quantizing the features of node $i$, the feature with the largest absolute value $f_i$ in the node features $x_i$ is first selected, and then we find the nearest $q_{max}$ and quantize the node features with the $(s, b)$ corresponding to this $q_{max}$. When performing backpropagation, we first calculate the gradients of the loss function w.r.t. quantization parameters according to Eq. 3 and Eq. 4. For a specific set of quantization parameters $(s_j, b_j)$, we collect the gradients from the nodes that have used them

---

**Algorithm 1** Nearest Neighbor Strategy

---

1: **ForwardPass** $(\boldsymbol{X} = (\boldsymbol{x}_1, \boldsymbol{x}_2, ..., \boldsymbol{x}_N)^T)$:
2:       Initialize$(\boldsymbol{s}, \boldsymbol{b}), \boldsymbol{s} \in \mathbb{R}_+^{m \times 1}, \boldsymbol{b} \in \mathbb{R}_+^{m \times 1}$ before training
3:       Calculate $\boldsymbol{q}_{max} = \boldsymbol{s} \odot (2^{\boldsymbol{b}-1} - 1)$
4:       Calculate the maximum absolute value in the features of each node: $f_i = \max_j abs(\boldsymbol{x}_i^{(j)})$
5:       Search the index of quantization parameters for each node: $index_i = \arg\min_k |f_i - q_{max}^k|$
6:       Quantize the $i$-th node features using $(s_{index_i}, b_{index_i})$
7:       return $\boldsymbol{X}_q$
8: **end**

---

Table 1: The results comparison on node-level tasks. The average bits are counted for each task when the best results are achieved.

| Dataset | Model | Accuracy | Average bits | Compression Ratio | Speedup |
|---------|-------|----------|--------------|-------------------|---------|
| **Cora** | GCN(FP32) | 81.5±0.7% | 32 | 1x | — |
| | GCN(DQ ) | 78.3±1.7% | 4 | 8x | 1x |
| | GCN(ours) | **80.9±0.6%** | **1.70** | **18.6x** | **2.00x** |
| | GIN(FP32) | 77.6±1.1% | 32 | 1x | — |
| | GIN(DQ ) | 69.9±3.4% | 4 | 8x | 1x |
| | GIN(ours) | **77.8±1.6%** | **2.37** | **13.4x** | **1.41x** |
| | GAT(FP32) | 83.1±0.4% | 32 | 1x | — |
| | GAT(DQ ) | 71.2±2.9% | 4 | 8x | 1x |
| | GAT(ours) | **82.6±0.6%** | **2.03** | **15.4x** | **1.49x** |
| **CiteSeer** | GCN(FP32) | 71.1±0.7% | 32 | 1x | — |
| | GCN(DQ ) | 66.9±2.4% | 4 | 8x | 1x |
| | GCN(ours) | **70.6±1.1%** | **1.87** | **17.0x** | **1.91x** |
| | GIN(FP32) | 66.1±0.9% | 32 | 1x | — |
| | GIN(DQ ) | 60.8±2.1% | 4 | 8x | 1x |
| | GIN(ours) | **65.1±1.7%** | **2.54** | **12.6x** | **1.37x** |
| | GAT(FP32) | 72.5±0.7% | 32 | 1x | — |
| | GAT(DQ ) | 67.6±1.5% | 4 | 8x | 1x |
| | GAT(ours) | **71.9±0.7%** | **1.94** | **16.2x** | **1.45x** |
| **PubMed** | GAT(FP32) | 79.0±0.3% | 32 | 1x | — |
| | GAT(DQ) | 70.6±12.5% | 4 | 8x | 1x |
| | GAT(ours) | **78.8±0.4%** | **2.12** | **15.1x** | **1.38x** |
| **ogbn-arxiv** | GCN(FP32) | 71.7±0.3% | 32 | 1x | — |
| | GCN(DQ) | 65.4±3.9% | 4 | 8x | 1x |
| | GCN(ours) | **71.1±0.3%** | **2.65** | **12.1x** | **1.28x** |

and add these gradients together. After the model has been trained, we obtain the quantization parameters $(\boldsymbol{s}, \boldsymbol{b})$. Since $q_{max}$ can be calculated and sorted in advance, searching the nearest $q_{max}$ can be implemented by binary searching. Usually, we set $m = 1000$ for all graph-level tasks in our paper and the overhead introduced to inference time is negligible.

## 4 EXPERIMENTS

### 4.1 EXPERIMENTAL SETTINGS

In this section, we evaluate our method on three typical GNN models, i.e., GCN, GIN, and GAT. And we compare our method with the FP32 GNN model and DQ-INT4 (Tailor et al., 2020) on eight datasets, including four node-level semi-learning tasks (Cora, CiteSeer, PubMed, ogbn-arxiv) (Hu et al., 2020; Yang et al., 2016) and four graph-level tasks (REDDIT-BINARY, MNIST, CIFAR10, ZINC) (Yanardag & Vishwanathan, 2015; Dwivedi et al., 2020), to demonstrate the generality and

Table 2: The results comparison on graph-level tasks.

| Dataset | Model | Accuracy (Loss↓) | Average bits | Compression ratio | Speedup |
|---|---|---|---|---|---|
| **MNIST** | GCN(FP32) | 90.1±0.2% | 32 | 1x | — |
| | GCN(DQ) | 84.4±1.3% | 4 | 8x | 1x |
| | GCN(ours) | **89.9±0.8%** | **3.50** | **9.12x** | **1.17x** |
| | GIN(FP32) | 96.4±0.4% | 32 | 1x | — |
| | GIN(DQ) | 95.5±0.4% | 4 | 8x | 1x |
| | GIN(ours) | **95.7±0.2%** | **3.75** | **8.52x** | **1.07x** |
| | GAT(FP32) | 95.6±0.1% | 32 | 1x | — |
| | GAT(DQ) | 93.1±0.3% | 4 | 8x | 1x |
| | GAT(ours) | **93.9±1.0%** | **3.86** | **8.28x** | **1.13x** |
| **CIFAR10** | GCN(FP32) | 55.9±0.4% | 32 | 1x | — |
| | GCN(DQ) | 51.1±0.7% | 4 | 8x | 1x |
| | GCN(ours) | **52.5±0.8%** | **3.32** | **9.62x** | **1.25x** |
| | GIN(FP32) | 57.5±0.7% | 32 | 1x | — |
| | GIN(DQ) | 50.7±1.6% | 4 | 8x | 1x |
| | GIN(ours) | **54.9±1.5%** | **3.53** | **9.06x** | **1.14x** |
| | GAT(FP32) | 65.4±0.4% | 32 | 1x | — |
| | GAT(DQ) | 56.5±0.6% | 4 | 8x | 1x |
| | GAT(ours) | **64.7±2.8%** | **3.73** | **8.57x** | **1.12x** |
| **ZINC** | GCN(FP32) | 0.450±0.008 | 32 | 1x | — |
| | GCN(DQ) | 0.536±0.011 | 4 | 8x | 1x |
| | GCN(ours) | **0.492±0.056** | **3.68** | **8.68x** | **1.08x** |
| **REDDIT-BINARY** | GIN(FP32) | 92.2±2.3% | 32 | 1x | — |
| | GIN(DQ) | 81.3±4.4% | 4 | 8x | 1x |
| | GIN(ours) | **90.8±1.8%** | **3.50** | **9.14x** | **1.16x** |

robustness of our method. Among these datasets, ZINC is a dataset for regression tasks, which uses regression loss as the metric of the model performance, while others are all for classification tasks.

For a fair comparison, we set the quantization bitwidth of $W$ for all GNNs to 4bits as DQ-INT4. We count the average bitwidths for nodes features in all layers of the overall model and list them in our results, denoted by "Average bits". Since today's CPUs and GPUs can not support mixed-precision operations well, we implement a precision-scalable hardware accelerator to perform the overall inference process for GNN. The accelerator employs massive bit-serial multipliers Judd et al. (2016), therefore, the latency of the integer multiplications is determined by the bitwidth of the node features. To evaluate the performance gains of our method over DQ-INT4, we develop a cycle-accurate simulator for our accelerator. More details about accelerator architecture are shown in Appendix A.7.5. Moreover, we show the compression ratio of quantized GNNs compared to the FP32 models in terms of overall memory size. For simplicity, we use GNN(DQ) to represent the GNNs quantized by DQ-INT4 and GNN-dataset to represent the task in which we run the experiment, e.g., GCN-Cora represents the GCN model trained on Cora. Detailed information about datasets and settings is in Appendix A.5 and Appendix A.6.

## 4.2 NODE-LEVEL TASKS

Table 1 shows the experimental results on three GNN architectures trained on four node-level datasets. Compared with DQ-INT4, our method can achieve significantly better accuracy on each task, even with a higher compression ratio, improving the inference performance with 1.28x to 2.00x speedups. On almost all node-level tasks, our proposed $A^2Q$ has negligible accuracy drop compared to the FP32 baselines while achieving 12.1x-18.6x compression ratio. Since both GIN and GAT involve more complex computations, such as the calculation of attention coefficients in GAT, it is more challenging to quantize those models, and DQ performs poorly on these two models. However, our method can overcome this problem and maintain comparable accuracy compared with the FP32 models. Our method can outperform the DQ-INT4 by 11.4% on the GAT-Cora task with a smaller bitwidth (2.03 v.s. 4). Even on ogbn-arxiv, which has a large number of nodes, $A^2Q$ can

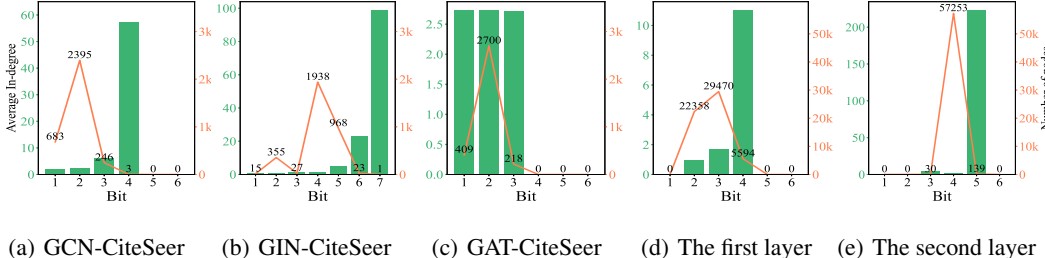

(a) GCN-CiteSeer    (b) GIN-CiteSeer    (c) GAT-CiteSeer    (d) The first layer    (e) The second layer

Figure 4: The relationship between quantized bitwidth and average in-degrees of nodes. (a), (b) and (c) represent the results of three GNN models trained on CiteSeer. (d) and (e) are results about the first and the second layer of an MLP, which is the update function of GIN trained on REDDIT-BINARY.

achieve a 12.1x compression ratio compared with FP32 baseline with comparable accuracy, which demonstrates the robustness of our method. Moreover, to demonstrate the generality of our method, we also evaluate our method on heterogeneous graphs and the inductive learning tasks and compare with more related works in Appendix A.7.1.

## 4.3  GRAPH-LEVEL TASKS

Table 2 presents the comparison results on the graph-level tasks. Our method can obtain better results on all tasks than DQ-INT4 with higher compression and a considerable speedup. Especially on the GIN-REDDIT-BINARY task, our method outperforms DQ-INT4 by 9.5%. Even for graph datasets with similar in-degrees, e.g., MNIST and CIFAR10, our method also learns the appropriate bitwidths for higher compression ratio and better accuracy. Although on GIN-MINST task, the improvement of our method is relatively small due to the similarity of the in-degrees between different nodes, our method can achieve comparable accuracy with smaller bitwidth (3.75 v.s. 4).

## 4.4  ANALYSIS

To understand why our approach works, we analyze the relationship between the learned bitwidths and the topology of the graph. Figure 4(a) and 4(b) reveal that the bitwidth learned by $A^2Q$ is strongly related to the topology of graph data in the node-level tasks. As the bitwidth increases, the average in-degrees of nodes become larger. In other words, $A^2Q$ method tends to learn higher bitwidth for nodes with higher in-degrees. However, in GAT, as shown in Figure 4(c), the learned bits are irregular. This is because the features aggregated in GAT are topology-free. However, our method can still learn appropriate quantization bitwidths for different nodes, which improves accuracy while reducing memory usage. In addition, Figure 4 also shows the node distribution for different bitwidths and the result is consistent with power-law distribution. Since nodes in graph data mainly have low in-degrees, most of the nodes are quantized to low bitwidth ($\leq 4$), compressing the GNNs as much as possible. And there are also some high in-degree nodes quantized to high bitwidth, which can help to maintain the accuracy of the GNN models. As a result, the average bitwidth of the entire graph features is low, and the accuracy degradation is negligible.

For the graph-level tasks in which the number of nodes varies, our method is also aggregation-aware. We select a layer of GIN trained on REDDIT-BINARY and analyze the relationship between bitwidth and average in-degrees of nodes using the corresponding bitwidth to quantize in Figure 4(d) and 4(e). It can be seen that the bitwidth learned for nodes features input to the second layer of MLP, which is the update function in GIN for graph-level tasks, does not present a correlation with the topology of graph. We analyze the reason and find that the node features before the second layer is the result mapped by the first layer of MLP and is activated by the activation function, e.g., ReLU, which results in the node features losing the topology information. We present more experiment results in Appendix A.7. to demonstrate that our method is generally applicable.

## 5  ABLATION STUDY

**The advantage of learning-based mixed-precision quantization:** In Figure 5, we compare our $A^2Q$ with the manual mixed-precision method, which manually assigns high-bit to those nodes with high in-degrees and low-bit to those nodes with low in-degrees. In the figure, the postfix "learn"

Table 3: Ablation Study.

| Model | Config | Accuracy | Average bits |
|---|---|---|---|
| **GIN-Cora** | no-lr | 33.7±4.1% | 4 |
| | no-lr-b | 75.6±0.2% | 4 |
| | no-lr-s | 56.1±4.9% | 3.85 |
| | lr-all | **77.8±1.6%** | **2.37** |
| **GCN-CiteSeer** | FP32 | 71.1±0.7% | 32 |
| | Global | 56.8±6.7% | 3 |
| | Local | **70.6±1.1%** | **1.87** |

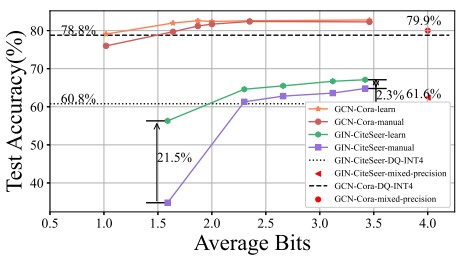

Figure 5: The comparison between learning bitwidth and assign manually.

denotes that using $A^2Q$ method, "manual" denotes that we assign bits to nodes and the model only learns the stepsize, and "mixed-precision" denotes that the model uses the same quantization method as DQ-INT4 but assigning different bitwidths to nodes. For the "mixed-precision", we assign 5bits to those nodes with 50% top in-degrees and assign 3bits to others. The implications are two-fold. First, compared with the DQ-INT4, which uses the same quantization bitwidth, the mixed-precision method obtains 1.1% gains on GCN-Cora tasks demonstrating that the mixed-precision method is more effective. Second, the results of the learning method outperform the manual method on all tasks. Especially for the models with a high compression ratio, on GIN-CiteSeer task, learning method can achieve 21.5% higher accuracy. This demonstrates that our learning method can perform better than the assignment method according to prior knowledge for mixed-precision quantization of GNNs.

**The power of learning the quantization parameters:** Ablations of two quantization parameters $(s, b)$ on the GIN-Cora task are reported in the first row of Table 3. The "no-lr" denotes that do not use learning method, "no-lr-b" denotes that only learn the step size $s$ , "no-lr-s" denotes that only learn the bitwidths $b$, and "lr-all" denotes that learn the bitwidth and step size simultaneously. We can see that learning the step size can significantly increase the accuracy and even the "no-lr-bit" model can outperform the DQ-INT4 at the same compression ratio. When learning the bitwidth and step size simultaneously, the model can achieve higher accuracy with a higher compression ratio. This is because our method learns lower bitwidths for most nodes with low in-degrees and higher bitwidths for a tiny fraction of nodes with high in-degrees, which can improve the compression ratio while achieving higher accuracy.

**Local Gradient v.s. Global Gradient:** To demonstrate the effectiveness of our Local Gradient method, we compare the models trained with and without it on the GCN-CiteSeer task in the last row of Table 3. The "Global" denotes that the model is trained with Eq. 3 and Eq. 4. The model trained with the local method outperforms the global method by 13.8% with a higher compression ratio. This is because the Local Gradient method can learn quantization parameters for all nodes, while only quantization parameters for a part of nodes can be updated with the Global Gradient method due to the extreme sparse connection in the graph on the node-level semi-supervised tasks.

**The overhead of Nearest Neighbor Strategy:** We evaluate the real inference time of the GIN model on the 2080ti GPU. On REDDIT-BINARY task, the model without the selection process requires 121.45ms, while it takes 122.60ms for the model with our Nearest Neighbor Strategy, which only introduces 0.95% overhead. But with the help of the Nearest Neighbor Strategy, our model can obtain 19.3% accuracy gains for quantized GIN on REDDIT-BINARY.

## 6 CONCLUSION

This paper proposes $A^2Q$, an aggregation-aware mixed-precision quantization method for GNNs, and introduces the Local Gradient and Nearest Neighbor Strategy to generalize $A^2Q$ to the node-level and graph-level tasks, respectively. Our method can learn the quantization parameters for different nodes by fully utilizing the property of GNNs and graph data. The model quantized by our $A^2Q$ can achieve up to a 18.6x compression ratio, and the accuracy degradation is negligible compared with the FP32 baseline. Compared with the prior SOTA, DQ-INT4, our method can significantly improve 11.4% accuracy with up to a 2.00x speedup on different tasks. Our work provides a general, robust and feasible solution to speed up the inference of GNNs.

ACKNOWLEDGMENTS

This work was supported in part by National Key Research and Development Program of China (No. 2020AAA0103400), the Strategic Priority Research Program of Chinese Academy of Sciences (No. XDB32050200), Jiangsu Key Research and Development Plan (No.BE2021012-2), and National Natural Science Foundation of China (No.62106267).

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

# A APPENDIX

## A.1 UNIFORM QUANTIZATION

In this section, we will give a detailed introduction to the content related to quantification.

### A.1.1 QUANTIZATION PROCESS

For a vector $\boldsymbol{x}$, the $\boldsymbol{x}_q$ is a quantized representation. Given the quantization step size $s$, $s \in \mathbb{R}_+$, and the quantization bitwidth $b$, $b \in \mathbb{N}_+$, then the uniform quantization is implemented as:

$$\bar{\boldsymbol{x}} = sign(\boldsymbol{x}) \begin{cases} \lfloor \frac{|\boldsymbol{x}|}{s} + 0.5 \rfloor, & |\boldsymbol{x}| < s(2^{b-1} - 1) \\ \\ 2^{b-1} - 1, & |\boldsymbol{x}| \geq s(2^{b-1} - 1) \end{cases} . \tag{9}$$

The $\boldsymbol{x}$ at 32bits is mapped to the integer number set $\{-2^{b-1} + 1, ..., 0, ..., 2^{b-1} - 1\}$ where the bitwidth is #$b$ bits, and the quantized representation can be calculated as $\boldsymbol{x}_q = s \cdot \bar{\boldsymbol{x}}$. For inference, $\bar{\boldsymbol{x}}$ can be used to compute matrix multiplication in the update phase or perform other computations in GNNs layers and the output of these computations then are rescaled by the corresponding $s$ using a relatively lower cost scalar-vector multiplication. As an illustrative example, for vectors $\boldsymbol{x} \in \mathbb{R}^{3 \times 1}$ and $\boldsymbol{y} \in \mathbb{R}^{3 \times 1}$, the quantization parameters are both $s = 0.1$, $b = 5$, the process of inner product between these two vectors by integers is shown in Figure 6. When the values in a vector are all non-negative, we do not need to represent the sign bit in the fixed-point representation. Therefore, the value can use #$b$ bits to quantize instead of using the first bit to represent the sign bit. Then the quantization range of uniform quantization is $[-s(2^b - 1), s(2^b - 1)]$.

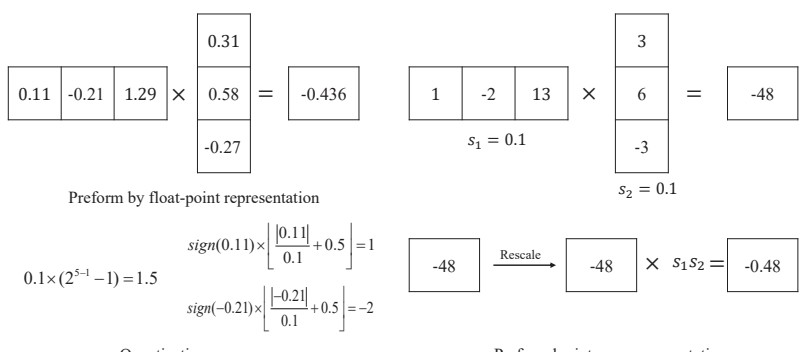

Figure 6: An example of performing inner product by integers representation.

### A.1.2 GRADIENT IN BACKPROPAGATION

Due to the floor function used in the quantization process is not differentiable, the gradient of $x_q$ with respect to $x$ vanishes almost everywhere, which makes it impossible to train the model by the backpropagation algorithm. Therefore, we use the straight-through estimator (Bengio et al., 2013) to approximate the gradient through the floor function, i.e., $\frac{\partial L}{\partial x} = \frac{\partial L}{\partial x_q} \mathbb{I}_{|x| \leq s(2^b - 1)}$, where $\mathbb{I}_{|x| \leq s(2^b - 1)}$ is a indicator function, whose value is 1 when $|x| \leq s(2^b - 1)$, and vice versa. In our paper, the quantification parameters $(s, b)$ are learnable, the gradients of $x_q$ w.r.t. $(s, b)$ used in Eq. 3 and Eq. 4 are:

$$\begin{bmatrix} \frac{\partial x_q}{\partial s} \\ \\ \frac{\partial x_q}{\partial b} \end{bmatrix} = \begin{cases} \begin{bmatrix} \frac{1}{s}(x_q - x) \\ 0 \end{bmatrix}, & |\boldsymbol{x}| < s(2^{b-1} - 1) \\ \\ sign(x) \begin{bmatrix} (2^{b-1} - 1) \\ 2^{b-1} \ln(2) s \end{bmatrix}, & |\boldsymbol{x}| \geq s(2^{b-1} - 1) \end{cases} . \tag{10}$$

Table 4: The aggregation functions and update functions for GNNs used in this paper, $d_i$ denotes the degree of node $i$, the $\varepsilon$ denotes a learnable constant, and $\alpha$ represent attention coefficients.

| Model | Aggregation function | Update function |
|-------|---------------------|-----------------|
| GCN | $\boldsymbol{h}_i^{(l)} = \sum\limits_{j \in \mathcal{N}(i) \cup \{i\}} \frac{1}{\sqrt{d_i}\sqrt{d_j}} \boldsymbol{x}_j^{(l-1)}$ | $\boldsymbol{x}_i^{(l)} = ReLU(\boldsymbol{W}^{(l)}\boldsymbol{h}_i^{(l)} + \boldsymbol{b}^{(l)})$ |
| GIN | $\boldsymbol{h}_i^{(l)} = (1 + \varepsilon^{(l)})\boldsymbol{x}_i^{(l-1)} + \sum\limits_{j \in \mathcal{N}(i)} \boldsymbol{x}_j^{(l-1)}$ | $\boldsymbol{x}_i^{(l)} = MLP^{(l)}(\boldsymbol{h}_i^{(l)}, \boldsymbol{W}^{(l)}, \boldsymbol{b}^{(l)})$ |
| GAT | $\boldsymbol{h}_i^{(l)} = \sum\limits_{j \in \mathcal{N}(i) \cup \{i\}} \alpha_{i,j}^{(l)} \boldsymbol{x}_j^{(l-1)}$ | $\boldsymbol{x}_i^{(l)} = \boldsymbol{W}^{(l)}\boldsymbol{h}_i^l + \boldsymbol{b}^{(l)}$ |

Table 5: The statistics for density of adjacency matrix and the labeled nodes in four node-level datasets.

|  | Cora | CiteSeer | PubMed | ogbn-arxiv |
|--|------|----------|--------|------------|
| Density of A | 0.144% | 0.112% | 0.028% | 0.008% |
| Labled nodes | 5.17% | 3.61% | 0.30% | 53.70% |

### A.2 MORE ABOUT GRAPH NEURAL NETWORKS

In this section, we first give detailed information about the MPNN framework (Gilmer et al., 2017), and then provide a detailed examination of the three GNNs used in our papers.

A graph $\mathcal{G} = (\mathcal{V}, \mathcal{E})$ consist of nodes $\mathcal{V} = \{1, ..., N\}$ and edges $\mathcal{E} \subseteq \mathcal{V} \times \mathcal{V}$ has node features $\boldsymbol{X} \in \mathbb{R}^{N \times F}$ and optionally H-dimensional edge features $\boldsymbol{E} \in \mathbb{R}^{E \times H}$. The MPNN framework can be formulated by $\boldsymbol{x}_i^{(l)} = \gamma^{(l)}(\boldsymbol{x}_i^{(l-1)}, \square_{j \in \mathcal{N}(i)} \phi^{(l)}(\boldsymbol{x}_i^{(l-1)}, \boldsymbol{x}_j^{(l-1)}, \boldsymbol{e}_{ij}^{(l-1)}))$, where $\phi$ is a differentiable kernel function, $\square$ is the aggregation function which is permutation-invariant, and the $\gamma$ is a learnable update function, $\boldsymbol{x}_i$ is the features of node $i$ and $\boldsymbol{e}_{ij}$ is the features of edge between node $i$ and $j$, $\mathcal{N}(i) = \{j : (i, j) \in \mathcal{E}\}$, and $l$ represents the $l$-th layer of the GNNs.

In this paper, we focus on three typical GNN models whose forwardpass all can be represented by the MPNN framework, Graph Convolution Network (GCN) (Kipf & Welling, 2016), Graph Isomorphism Network (GIN) (Xu et al., 2018), and Graph Attention Network (GAT) (Veličković et al., 2017). the detailed information is shown in Table 4.

### A.3 PROOFS OF THEORETICAL RESULTS

This section provides formal proof of the theoretical results of our paper.

#### A.3.1 NOTATIONS

Here, we define the notations utilized in our proof. $A = \{0, 1\}^{N \times N}$ is the adjacency matrix that indicates whether there is an edge between each pair of nodes, e.g., if there is an edge between node $i$ and node $j$, then $a_{ij} = 1$, otherwise, $a_{ij} = 0$. Then, $\tilde{A} = A + I$ is the adjacency matrix for a graph

that is added to the self-loops. The degree matrix $D = diag(d_1, d_2, ..., d_n)$, where $d_i = \sum_j a_{ij}$ and the degree matrix for the graph having self-loops is $\tilde{D} = (\tilde{d}_1, \tilde{d}_2, ..., \tilde{d}_n)$, where $\tilde{d}_i = \sum_j \tilde{a}_{ij}$.

### A.3.2 PROOFS

**Proof 1.** *The gradients of the loss function with respect to the node features in semi-supervised tasks are most zero.*

Without loss of generality, we use the GCN model as an example. From the Table 4, the graph convolution operation can be described as

$$X^{(l+1)} = \sigma(\hat{A} X^{(l)} W^{(l)}), \tag{11}$$

where $\hat{A} = \tilde{D}^{-\frac{1}{2}} \tilde{A} \tilde{D}^{-\frac{1}{2}}$, is the normalized adjacency matrix, $W^{(l)} \in \mathbb{R}^{F_{in} \times F_{out}}$ is a learnable weight matrix in the $l$-th layer of GCN. $X^{(l)}$ is the input of the $l$-th layer and the output of the $(l-1)$-th layer in GCN. $\sigma$ is the non-linear activation function, e.g., ReLU. Note that the $\hat{A}$ is an extreme sparse matrix for node-level datasets in our paper.

In our training process of the model, we use $nll\_loss$ as our task loss function $L$. Only the nodes in the train set $T$ have labels. For the last layer of GCN, we get the node features to be classified by $H^{(l+1)} = softmax(X^{(l+1)})$. Then the gradient of $L$ with respect to $X^{(l+1)}$ is

$$G^1 = \nabla_{X^{(l+1)}} L = \frac{\partial L}{\partial H^{(l+1)}} \cdot \frac{\partial H^{(l+1)}}{\partial X^{(l+1)}} = [l_{ij}] \in \mathbb{R}^{N \times F_{out}}, \tag{12}$$

where only the $G^1_{i,:}$, $i \in T$ is not zero, otherwise, $G^1_{i,:} = 0$. Then, the gradient of the loss function with respect to $X^{(l)}$ is

$$G^2 = \nabla_{X^{(l)}} L = \hat{A}^T (\nabla_{X^{(l+1)}} L \odot \sigma'(\hat{A} X^{(l)} W^{(l)}))(W^{(l)})^T. \tag{13}$$

For node $j$ do not have an edge with the node $i$, $i \in T$, $G^2_{j,:} = 0$. Table 5 lists the density of the adjacency matrix $A$ and the percentage of the labeled nodes in four node-level datasets. Because the sparsity property of adjacency matrix and the nodes with trained labels only account for a tiny fraction of the graph, the gradients from the loss function for most node features are zero.

**Proof 2.** *The normalized adjacency matrix $\hat{A}$ is not needed to be quantized for the GCN model.*

We take the process of $XW \rightarrow A(XW)$ as an illustrative example, which represents first calculate the $B = XW$ and then calculate $AB$. For the $l$-th layer of FP32 models, the first stage is $B_l = X_l W_l$, and then calculate the $X_{l+1} = \hat{A} B_l$, where $X_l \in \mathbb{R}^{N \times F_1}$, $W_l \in \mathbb{R}^{F_1 \times F_2}$ and $A \in \mathbb{R}^{N \times N}$. The step-size for $B_l$, $X_l$ and $W_l$ is $S_{B_l}$, $S_{X_l}$ and $S_{W_l}$, respectively. And they are all diagonal matrices. The integer representations are calculated as $B_l = B_{l\_q} S_{B_l}$, $X_l = S_{X_l} X_{l\_q}$ and $W_l = W_{l\_q} S_{W_l}$. Note that for the node-level tasks, we can obtain the $S_{B_l}$, $S_{X_l}$ and $S_{W_l}$ in advance. And for the graph-level tasks, we can obtain them through one more element-wise multiplication whose overhead is negligible, as the comparison in Table 6. Then the first stage is:

$$\begin{aligned} B_l &= X_l \cdot W_l \\ &= (S_{X_l} \cdot X_{l\_q}) \cdot (W_{l\_q} \cdot S_{W_l}) \end{aligned}, \tag{14}$$

and there exists $B_l = B_{l\_q} S_{B_l}$. Therefore, the integers representation for the next stage can be calculated as:

$$\begin{aligned} B_{l\_q} &= B_l S_{B_l}^{-1} \\ &= (S_{X_l} \cdot X_{l\_q}) \cdot (W_{l\_q} \cdot S_{W_l}) S_{B_l}^{-1} \\ &= (S_{X_l} \cdot X_{l\_q}) \cdot (W_{l\_q} \cdot (S_{W_l} S_{B_l}^{-1})) \\ &= (S_{X_l} \otimes (S_{W_l} S_{B_l}^{-1})) \odot (X_{l\_q} \cdot W_{l\_q}) \end{aligned}, \tag{15}$$

where the $(S_{X_l} \otimes (S_{W_l} S_{B_l}^{-1}))$ can be calculated offline. Then we obtain the fixed-point representation $B_{l\_q}$ for the next stage and do not introduce overhead.

The process of node degree normalization after the aggregation process can be represented as $X_{l+1} = \sigma(\hat{A} B_l)$, where $\hat{A} = D^{-\frac{1}{2}} \tilde{A} D^{-\frac{1}{2}}$ is the normalized adjacency matrix, and $\sigma$ is the

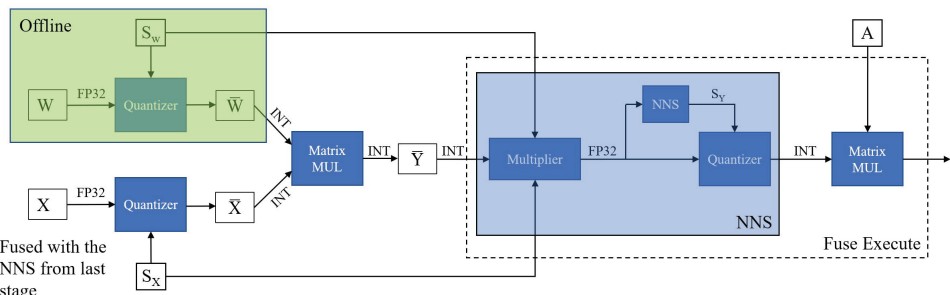

Figure 7: The pipeline of the quantization process on our accelerator.

non-linear activation function. $\boldsymbol{D}^{-\frac{1}{2}}$ at the right side of $\tilde{\boldsymbol{A}}$ can be fused into the $\boldsymbol{S}_{X_l}$ and then calculate $\boldsymbol{B}_{l\_q}$ as Eq. 15.

Then the features of the $(l+1)$-th layer $\boldsymbol{X}_{l+1}$ can be obtained as $\boldsymbol{X}_{l+1} = \sigma(\boldsymbol{D}^{-\frac{1}{2}}\tilde{\boldsymbol{A}}\boldsymbol{B}_{l\_q})$. And there exits $\boldsymbol{X}_{l+1} = \boldsymbol{S}_{X_{l+1}}\boldsymbol{X}_{(l+1)\_q}$. Therefore, the $\boldsymbol{X}_{(l+1)\_q}$ can be obtained as:

$$
\begin{aligned}
\boldsymbol{X}_{(l+1)\_q} &= \boldsymbol{S}_{X_{l+1}}^{-1}\boldsymbol{X}_{l+1} \\
&= \boldsymbol{S}_{X_{l+1}}^{-1}\sigma(\boldsymbol{D}^{-\frac{1}{2}}\tilde{\boldsymbol{A}}\boldsymbol{B}_{l\_q})
\end{aligned}
\tag{16}
$$

Note that the elements in diagonal matrix $\boldsymbol{S}_{X_{l+1}}$ are all positive because this matrix is made up of step-size, which is always positive. Then we can obtain $\boldsymbol{X}_{(l+1)\_q} = \sigma(\boldsymbol{S}_{X_{l+1}}^{-1}\boldsymbol{D}^{-\frac{1}{2}}\tilde{\boldsymbol{A}}\boldsymbol{B}_{l\_q})$, where $\boldsymbol{S}_{X_{l+1}}^{-1}\boldsymbol{D}^{-\frac{1}{2}}$ can be obtained before inference and $\tilde{\boldsymbol{A}} \in \{0, 1\}^{N \times N}$. The computation of $\tilde{\boldsymbol{A}}\boldsymbol{B}_{l\_q}$ only has addition operations and the $\boldsymbol{S}_{X_{l+1}}^{-1}\boldsymbol{D}^{-\frac{1}{2}}$ can be obtained before inference for node-level tasks or introduce only once more element-wise multiplication to calculate for the graph-level tasks.

The $\boldsymbol{D}^{-\frac{1}{2}}$ at the left side is fused into the element-wise multiplication performed by the next layer and the $\boldsymbol{D}^{-\frac{1}{2}}$ at the right side is fused into the element-wise multiplication performed by the current layer and the element-wise multiplication is a necessary stage in the quantized model. Therefore, we can perform the node degree normalization using fixed point addition operation instead of quantizing the normalized adjacency matrix which may introduce more quantization error.

**Proof 3.** *The quantization process can be fused with Batch Normalization operations.*

When GNNs have Batch Normalization (BN) Layers, the calculation process is as follows (Note that we have fused the mean and standard-deviation with the learned parameters in BN):

$$
\begin{aligned}
\boldsymbol{X}_{l+1} &= BN(\sigma(\hat{\boldsymbol{A}}\boldsymbol{B}_{l\_q})) \\
&= \sigma(\hat{\boldsymbol{A}}\boldsymbol{B}_{l\_q})\boldsymbol{Y} + \boldsymbol{Z}
\end{aligned}
\tag{17}
$$

where $\boldsymbol{Y} = diag(y_1, y_2, ..., y_{F_2}) \in \mathbb{R}^{F_2 \times F_2}$, $\boldsymbol{Z} = (\boldsymbol{z}_1, \boldsymbol{z}_2, ..., \boldsymbol{z}_{F_2}) \in \mathbb{R}^{N \times F_2}$ and $\boldsymbol{z}_i = (\theta_i, \theta_i, ..., \theta_i)^T \in \mathbb{R}^N$ among which $y_i$ and $\theta_i$ are the BN parameters for the i-th dimension feature of the nodes features. And there exits that $\boldsymbol{X}_{l+1} = \boldsymbol{S}_{X_{l+1}}\boldsymbol{X}_{l+1\_q}$. Therefore,

$$
\begin{aligned}
\boldsymbol{X}_{l+1\_q} &= \boldsymbol{S}_{X_{l+1}}^{-1}\boldsymbol{X}_{l+1} \\
&= \boldsymbol{S}_{X_{l+1}}^{-1}(\sigma(\hat{\boldsymbol{A}}\boldsymbol{B}_{l\_q})\boldsymbol{Y} + \boldsymbol{Z}) \\
&= (\boldsymbol{S}_{X_{l+1}}^{-1} \otimes \boldsymbol{Y}) \odot (\sigma(\hat{\boldsymbol{A}}\boldsymbol{B}_{l\_q})) + \boldsymbol{S}_{X_{l+1}}^{-1}\boldsymbol{Z}
\end{aligned}
\tag{18}
$$

Through Eq. 18, we can fuse the quantization of the next layer into the BN operation of the current layer, which will not introduce overhead because the BN layer itself requires floating point operations. Note that the float point operations are also element-wise.

Table 6: The comparison between fixed-point operations and float-point operations for some tasks using the Nearest Neighbor Strategy.

| Task | GIN-RE-IB | GCN-MNIST | GAT-CIFAR10 | GCN-ZINC |
|---|---|---|---|---|
| Fixed-point(M) | 936.96 | 455.69 | 1387.98 | 504.62 |
| Float-point(M) | 7.35 | 2.06 | 13.71 | 1.74 |
| Ratio | 0.78% | 0.45% | 0.98% | 0.34% |

### A.4 THE OVERHEAD ANALYSIS OF NEAREST NEIGHBOR STRATEGY

Through our dedicated hardware and the optimized pipeline, we reduce the overhead introduced by the Nearest Neighbor Strategy (NNS) as much as possible. As the pipeline is shown in Figure 7, we fuse the (NNS) with the following operations. The fixed-point results produced by the previous stage are used to first multiply the corresponding step-size from the previous stage (an element-wise float point multiplication) and then execute the NNS process. After getting the step-size, these features are quantized immediately (an element-wise float point multiplication). Therefore, through this fusion way, we do not need the extra memory to store a copy of FP32 features.

In addition, the overhead of the NNS is from one more element-wise float point multiplication and the search process. We provide a comparison of the number of float-point operations and fixed-point operations for different graph-level tasks in Table 6, where 'Fixed-point' denotes the fixed-point operation, 'Float-point' denotes the float-point operation and the 'Ratio' denotes the percentage of the float-point operations in the overall process. The extra float-point operations introduced by NNS is only a tiny fraction of the fixed-point operations. On the other hand, through our optimized pipeline and the comparator array used in our accelerator the latency introduced by the search process of the NNS can be overlapped. Therefore, the overhead introduced by NNS is negligible.

### A.5 DATASETS

We show the statistics for each dataset used in our work in Table 7. For datasets in node-level tasks, nodes correspond to documents and edges to citations between them. Node features are a bag-of-words representation of the document. The target is to classify each node in the graph correctly. The **Cora**, **CiteSeer** and **PubMed** are from Yang et al. (2016). The **ogbn-arxiv**, **ogbl-mag** and **ogbn-collab** are from Hu et al. (2020). The **Flickr** is from Zeng et al. (2019). The **Reddit** is from Hamilton et al. (2017). In graph-level tasks, **REDDIT-BINARY** (Yanardag & Vishwanathan, 2015) is a balanced dataset where each graph corresponds to an online discussion thread and the nodes correspond to users. There would be an edge between two nodes if at least one of them responded to another's comment. The task is then to identify whether a given graph belongs to a question/answer-based community or a discussion-based community The **MNIST** and **CIFAR-10** datasets (Dwivedi et al., 2020) which are often used for image classification tasks are transformed into graphs in which every node is represented by their superpixel and location, and the edges are constructed by Achanta et al. (2012). The task is to classify the image using its graph representation. The **ZINC** Gómez-Bombarelli et al. (2018) dataset contains graphs representing molecules, where each node is an atom. The task is to regress the penalized $\log$P (also called constrained solubility in some works) of a given graph. In Figure 8, we show the in-degree distribution for all the datasets we use in our paper.

### A.6 EXPERIMENTAL SETUP

To make a fair comparison, we adopt the same GNN architectures as Tailor et al. (2020) on every task, and the FP32 baseline is also the same. For those tasks that Tailor et al. (2020) does not do, we adopt the same architecture as their FP32 version. For **ogbn-arxiv** and **PubMed**, we use the

Table 7: The statistics for each dataset used in this work.

| Task | Name | Graphs | Nodes | Edges | Features | Classes |
|---|---|---|---|---|---|---|
| Node-level | Cora | 1 | 2708 | 10556 | 1433 | 7 |
| | CiteSeer | 1 | 3327 | 9104 | 3703 | 6 |
| | PubMed | 1 | 19717 | 88648 | 500 | 3 |
| | ogbn-arxiv | 1 | 169343 | 1166243 | 128 | 23 |
| | ogbn-mag | 1 | 1939743 | 25582108 | 128 | 349 |
| | ogbl-collab | 1 | 235868 | 1285465 | 128 | – |
| | Reddit | 1 | 232965 | 11606919 | 602 | 41 |
| | Flickr | 1 | 89250 | 899756 | 500 | 7 |
| Graph-level | REDDIT-BINARY | 2000 | $\sim$429.6 | $\sim$995.5 | 0 | 2 |
| | MNIST | 70000 | $\sim$71 | $\sim$565 | 3 | 10 |
| | CIFAR10 | 60000 | $\sim$117.6 | $\sim$941.2 | 5 | 10 |
| | ZINC | 12000 | $\sim$23 | $\sim$49.8 | 28 | — |

architectures and FP32 results reported by Hu et al. (2020) and Kipf & Welling (2016) respectively. We use standard splits for MNIST, CIFAR-10, and ZINC (Dwivedi et al., 2020). For **Cora**, **CiteSeer** and **PubMed**, we use the splits used by Yang et al. (2016). For REDDIT-BINARY, we use 10-fold cross-validation. Our data split way is also the same as DQ-INT4.

Figure 9 shows the architectures of the models used in our evaluations, including the layers, the number of hidden units, and whether to use a skip connection.

Our method is implemented using PyTorch Geometric (Fey & Lenssen, 2019). We quantize the same parts as the DQ-INT4 in all models except for the normalized adjacency matrix in the GCN model, which we have proven that the quantization of this matrix is not necessary in Appendix A.3.2, **proof 2.**. The values in the **Cora** and **CiteSeer** are all 0 or 1, therefore, we do not quantize the input features for the first layer of the GNNs trained on the two datasets as DQ. For all quantized GNNs, we train them by Adam optimizer. The learning rate and the learning rate schedule are consistent with their FP32 version. In our method, the quantization parameters $(s, b)$ are also learnable, so we set the learning rate for them, including the $b$ for features, $s$ for features, and $s$ for weights.

When initializing, the parameters of the models are initialized as their FP32 version, the quantization bits for all nodes and weight matrixes are initialized by 4bits, and the step sizes for node features and weights are initialized by $s \in \mathcal{N}(0.01, 0.01)$ except for the graph-level tasks on GAT, where we initialize the step size by $s \in \mathcal{U}(0, 1)$. The $\mathcal{N}$ is normal distribution and the $\mathcal{U}$ is uniform distribution. And for GAT model trained on graph-level datasets, we just learn the quantization bits of the node features, while in the attention coefficients computation part, we use the exact 4bit to quantize. The batch size is 128 in all graph-level tasks. The results reported in our work for GNNs on **Cora**, **CiteSeer** and **PubMed** are averaged over 100 runs with different seeds, and the results for **ogbn-arxiv**, **MNIST**, **CIFAR-10** and **ZINC** are averaged over ten runs. The results on **REDDIT-BINARY** are obtained by 10-fold cross-validation and the split seed is 12345, which is the same as DQ-INT4. All experiments in our paper ran on RTX 2080Ti GPU driven by Ubuntu 18.04. The version of the CUDA and Pytorch are 10.2 and 1.8.0, respectively.

### A.6.1 EXPERIMENTAL SETUPS FOR THE ABLATION STUDY.

**The advantage of learning-based mixed-precision quantization:** During the experiment of comparing the learning bitwidth and bit assignment, we ensure the average bits of node features of these two methods are comparable to verify the effectiveness of our $A^2Q$ method. As an example, if the average bit is 2.2bit when assigning the bit to nodes with different in-degrees, we will first sort the

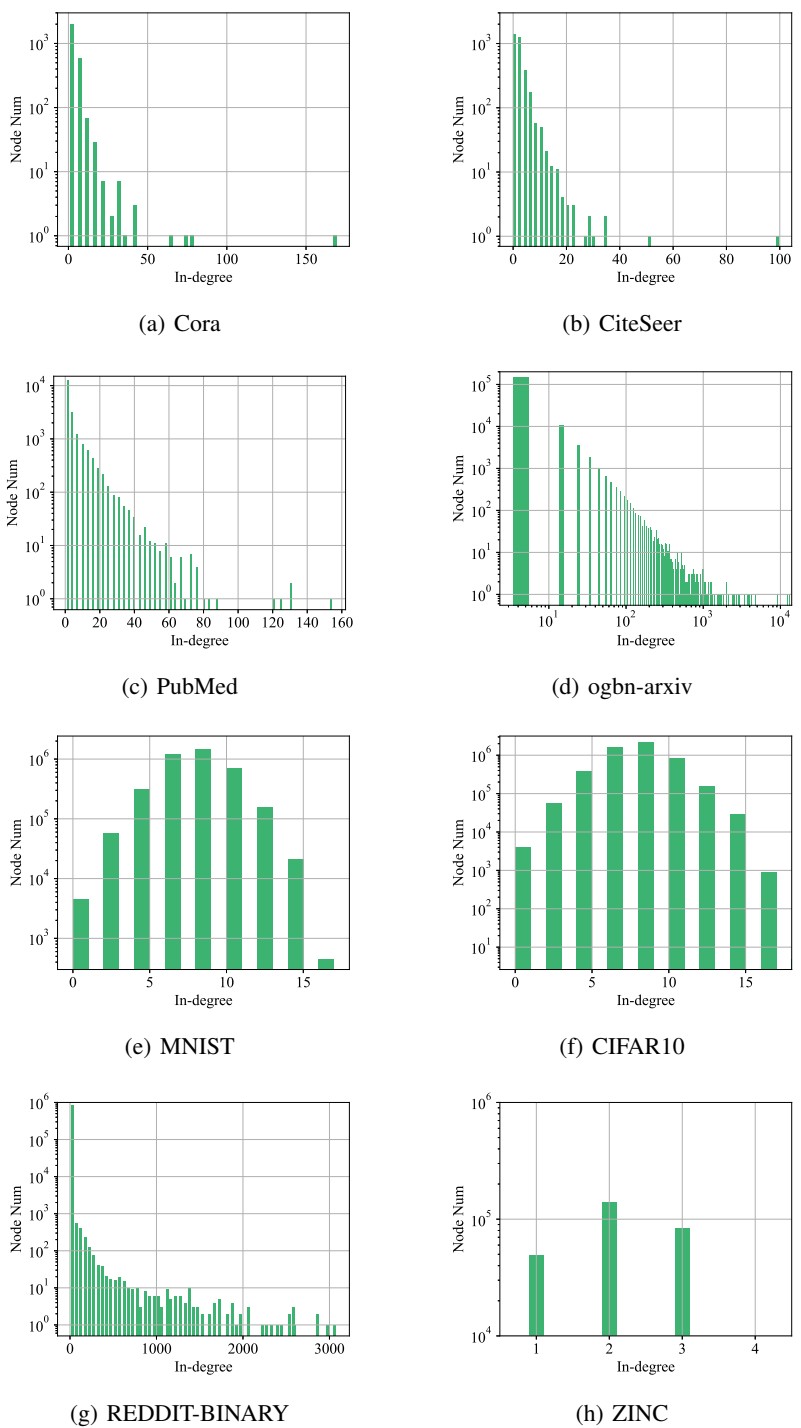

Figure 8: The in-degree distribution for each dataset used in this work.

nodes by their in-degrees and then select the nodes with the top 20% in-degrees, and quantize those by 3bit, and for the remaining nodes, we use 2bit to quantize. In the model trained by the bit assignment method, the bit is not learnable, and other hyperparameters are all consistent with the model using the $A^2Q$ method. For the "GCN-Cora-mixed-precision" and "GIN-CiteSeer-mixed-precision"

| Dataset | Archi | Cora | CiteSeer | PubMed | ogbn-arxiv | MNIST | CIFAR10 | ZINC | REDDIT-BINARY |
|---------|-------|------|----------|--------|------------|-------|---------|------|---------------|
| GCN | Layers | 2 | 2 | 2 | 3 | 5+1MLP | 5+1MLP | 5+1MLP | ---- |
| | Hidden units | 16/32/64 | 16/32/64 | 16/32/64 | 256 | 146 | 146 | 145 | ---- |
| | Skip connection | NO | NO | NO | NO | YES | YES | YES | ---- |
| GIN | Layers | 2 | 2 | 2 | 3 | 5+1MLP | 5+1MLP | 5+1MLP | 5 |
| | Hidden units | 16/32/64 | 16/32/64 | 16/32/64 | 256 | 110 | 110 | 110 | 64 |
| | Skip connection | NO | NO | NO | NO | YES | YES | YES | NO |
| GAT | Layers | 2 | 2 | 2 | ---- | 5+1MLP | 5+1MLP | 5+1MLP | ---- |
| | Hidden units | 8*head_numbers | 8*head_numbers | 8*head_numbers | ---- | 19*head_numbers | 19*head_numbers | 18*head_numbers | ---- |
| | Skip connection | NO | NO | NO | ---- | YES | YES | YES | ---- |

Figure 9: The model architectures used in our evaluations, the head number for all GAT models on different tasks are 8.

Table 8: The results comparison on GCN-PubMed and GIN-ogbn-arxiv.

| | | Accuracy | Average bits | Compression Ratio | Speedup |
|---|---|----------|--------------|-------------------|---------|
| **PubMed** | GCN(FP32) | 78.9±0.7% | 32 | 1x | — |
| | GCN(DQ) | 62.5±2.4% | 4 | 8x | 1x |
| | GCN(ours) | **77.5±0.1%** | **1.90** | **16.8x** | **1.45x** |
| **ogbn-arxiv** | GIN(FP32) | 68.8±0.2% | 32 | 1x | — |
| | GIN(DQ) | 57.6±2.2% | 4 | 8x | 1x |
| | GIN(ours) | **65.2±0.4%** | **3.82** | **8.4x** | **1.02x** |

tasks, we use 3bit and 5bit to quantize the GNNs while keeping the average bitwidth at 4bits. In particular, we assign 5bits to those nodes with 50% top in-degrees and assign 3bits to others.

**The power of learning the quantization parameters:** For the "no-lr-bit", we initialize the bitwidth as 4bits for all nodes features and just train the step size. For the "no-lr-step", we initialize the step size as previously mentioned but do not train them. For the "no-lr", we just initialize the bitwidth and the step size, but do not train them.

**Local Gradient v.s. Global Gradient:** All settings of the model trained by global gradient is consistent with the model trained by local gradient method.

**The overhead of Nearest Neighbor Strategy:** The model, without using the Nearest Neighbor Strategy, selects the quantization parameters according to their in-degrees. Every in-degree has a corresponding group of quantization parameters. Those nodes whose in-degrees are larger than 1000 will share the same group quantization parameters. In this way, The quantization parameters used by the nodes features can be determined as soon as the graph data is available, without the need for selection during the inference process, and then we can compare the overhead introduced by the selection process.

## A.7 MORE EXPERIMENTS RESULTS

This section is a complementary part about experiments results to demonstrate that our $A^2Q$ quantization method is general and robust.

Table 9: The results comparison on inductive learning tasks and more graphs.

| Task | Acc(%) | Average bits | Compression Ratio |
|---|---|---|---|
| GCN-mag | 30.8±0.1(FP32) | 32 | 1x |
| | 32.7±0.4(Ours) | 2.7 | 11.7x |
| GCN-collab | 44.8±1.1(FP32) | 32 | 1x |
| | 44.9±1.5(Ours) | 2.5 | 12.7x |
| GraphSage-REDDIT | 95.2±0.1(FP32) | 32 | 1x |
| | 95.3±0.1(Ours) | 3.9 | 8.1x |
| GraphSage-Flickr | 50.9±1.0(FP32) | 32 | 1x |
| | 50.0±0.5%(Ours) | 3.8 | 8.4x |

Table 10: Comparison with more quantization method.

| Task | Acc(%) | Average Bits | Compression Ratio |
|---|---|---|---|
| GCN-Cora | 80.9±0.0(Half-pre) | 16 | 1x |
| | 80.9±0.6(Ours) | 1.7 | 9.40x |
| GAT-CiteSeer | 68.0±0.1(LPGNAS) | 8 | 1x |
| | 71.9±0.7(Ours) | 1.9 | 4.21x |
| GraphSage-Cora | 74.3±0.1(LPGNAS) | 12 | 1x |
| | 74.5±0.2(Ours) | 2.7 | 4.44x |
| GraphSage-Flickr | 49.7±0.3(LPGNAS) | 8 | 1x |
| | 50.0±0.5(Ours) | 3.8 | 2.11x |

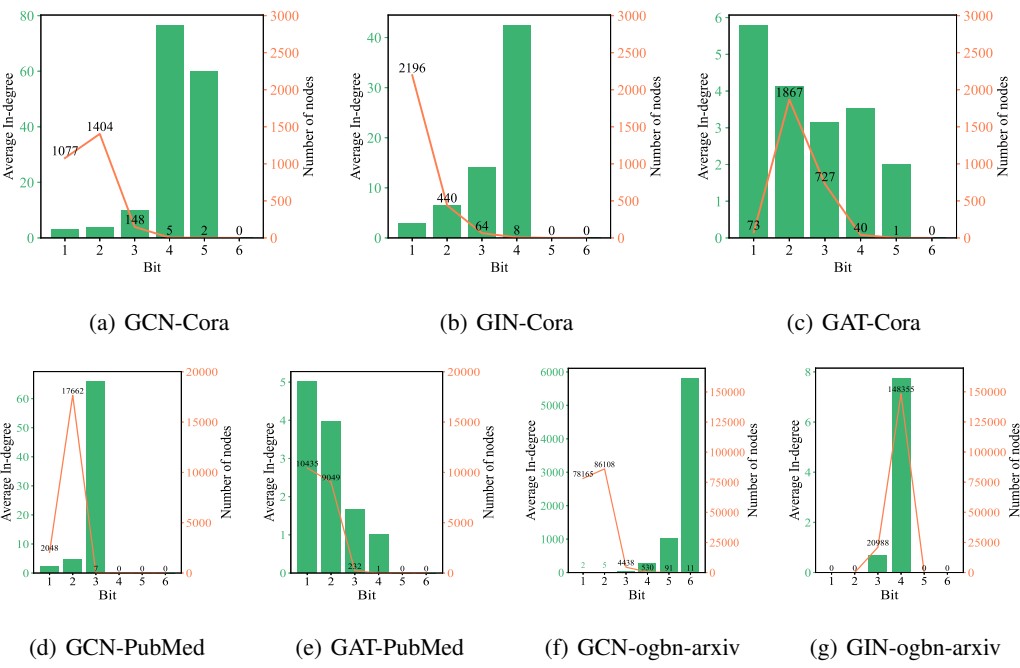

Figure 10: The relationship between bit and average in-degrees of nodes using the corresponding bitwidth to quantize. (a), (b) and (c) Three GNN models trained on Cora. (d) and (e) GCN and GAT trained on PubMed, respectively. (f) and (g) GCN and GIN trained on ogbn-arxiv, respectively.

Table 11: The effect of #m on the accuracy of quantized model, using GIN trained on REDDIT-BINARY as an example. The average bitwidth is 4bits.

| GIN(FP32): 92.2± 2.3% | | | GIN(DQ): 81.3± 4.4% | | |
|---|---|---|---|---|---|
| **m** | 100 | 400 | 800 | 1000 | 1500 |
| **Accuracy** | 88.7±3.5% | 90.6±3.8% | 92.0±2.2% | 92.5±1.8% | 92.6±1.9% |

### A.7.1 NODE-LEVEL TASKS

In Table 8, we show more task results on **PubMed** and **ogbn-arxiv**. On the GAT-ogbn-arxiv task, our GPU raised the Out Of Memory error, so we do not report the results on the GAT-ogbn-arxiv task. The model quantized by our $A^2Q$ method is also significantly better than DQ-INT4, which shows that our $A^2Q$ is general. We do not compare with DQ-INT8 because our results are comparable with the FP32 baseline with a much larger compression ratio than DQ-INT8.

We also show the relationship between bit and average in-degrees of nodes using the corresponding bitwidth to quantize on more tasks in Figure 10. We present the results of the final layer of GNNs. The results show that the bitwidth learned by our $A^2Q$ method is also aggregation-aware, which means that our method is robust.

We also evaluate the inductive model, GraphSage, on some other node-level tasks to demonstrate the generality of our method on inductive learning tasks. Due to the sampling operation in the GraphSage model, the subgraph input to the model varies, we apply our nearest neighbor strategy to these tasks, i.e., GraphSage-Flickr and GraphSage-Reddit. In addition, we evaluate our method on more datasets, such as the ogbn-mag and ogbl-collab. ogbn-mag is a heterogeneous graph and the ogbl-collab is used for the link prediction tasks.

The results of our experiments are presented in Table 9, where we can see that our approach still works well and even brings some generalization performance improvement while significantly compressing the model size. This also demonstrates that our Neighbor Nearest Strategy generalizes well on inductive models for node-level tasks.

We also compare with more quantization methods on GNNs. Zhao et al. (2020) uses the Network Architecture Search (NAS) to search for the best quantization strategy for different components in the GNNs. Brennan et al. (2020) explore the use of half-precision (i.e., FP16) in the forward and backward passes of GNNs. Table 10 presents the comparison results on various tasks with these two methods. 'Half-pre' denotes the method in Brennan et al. (2020), and 'LPGNAS' denotes the method in Zhao et al. (2020). The results demonstrate that our method achieves better accuracy with a smaller quantization bitwidth on all tasks.

### A.7.2 GRAPH-LEVEL TASKS

We propose the Nearest Neighbor Strategy to quantize the node features in graph-level tasks, in which the number of nodes input to models is various. In our Nearest Neighbor Strategy, $\#m$ groups quantization parameters $(s, b)$ should be initialized, and we explore the effect of the value of m on the performance of the quantized model in Table 11 using the GIN trained on REDDIT-BINARY dataset. We can observe that when the value of $m$ is smaller than 800, the accuracy increases as the value of $m$ increases. When the value of $m$ is higher than 800, the performances of the models with different $m$ are similar. However, the models with a larger $m$ are more stable.

Moreover, the selection of $m$ may be related to the number of nodes input to the model. According to our experiments, we finally select $m$ as 1000 for all graph-level tasks.

Table 12 lists the comparison results on GIN-ZINC and GAT-ZINC. On the regression tasks, our method is also significantly better than DQ-INT4. Notably, we do not learn different bitwidths for the nodes in ZINC datasets due to the similar topology structure between nodes.

Table 12: The results comparison on GIN-ZINC and GAT-ZINC.

| Modle | Dataset | Loss↓ | Average bits | Compression Ratio |
|---|---|---|---|---|
| | GAT(FP32) | 0.455±0.006 | 32 | 1x |
| | GAT(DQ) | 0.520±0.021 | 4 | 8x |
| **ZINC** | GAT(ours) | **0.495±0.006** | 4 | 8x |
| | GIN(FP32) | 0.334±0.024 | 32 | 1x |
| | GIN(DQ) | 0.431±0.012 | 4 | 8x |
| | GIN(ours) | **0.380±0.022** | 4 | 8x |

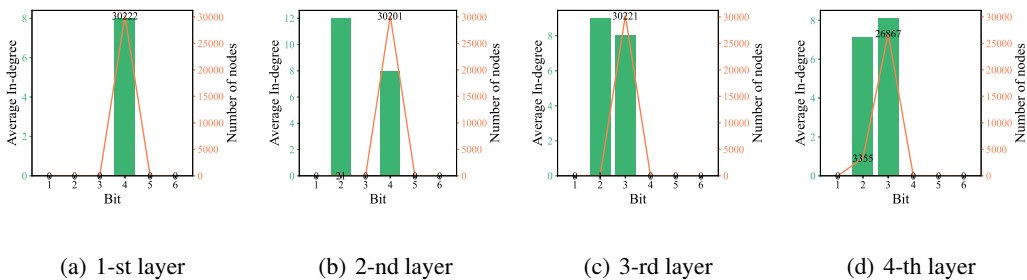

(a) 1-st layer      (b) 2-nd layer      (c) 3-rd layer      (d) 4-th layer

Figure 11: The relationship between bit and average in-degrees of nodes using the corresponding bitwidth to quantize on different layers of GCN trained on CIFAR10.

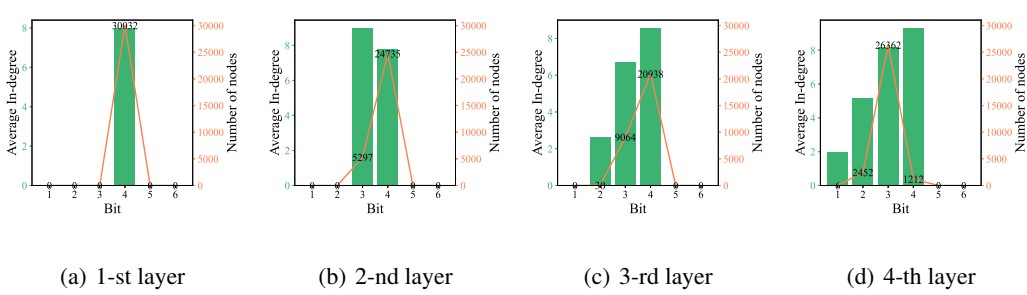

(a) 1-st layer      (b) 2-nd layer      (c) 3-rd layer      (d) 4-th layer

Figure 12: The relationship between bit and average in-degrees of nodes using the corresponding bitwidth to quantize on different layers of GIN trained on CIFAR10.

We also show the relationship between bit and average in-degree of nodes using the corresponding bit to quantize for more graph-level tasks in different layers immediately after the aggregation phase in Figure 11-Figure 16. The quantization bitwidths learned for graph-level tasks are also aggregation-aware. Because the difference of the in-degrees between different nodes is little in the MNIST and CIFAR10 dataset resulting in the aggregated features are similar between different nodes, the relationship between learned bitwidths and the in-degrees is irregular in some layers, e.g., the 2-nd layer in GCN trained on MNIST.

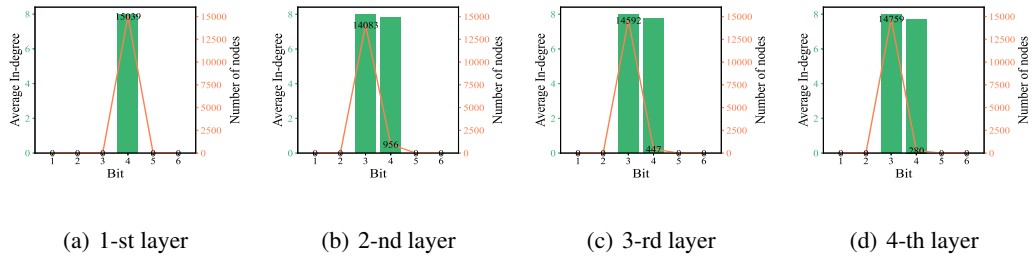

(a) 1-st layer    (b) 2-nd layer    (c) 3-rd layer    (d) 4-th layer

Figure 13: The relationship between bit and average in-degrees of nodes using the corresponding bitwidth to quantize on different layers of GAT trained on CIFAR10.

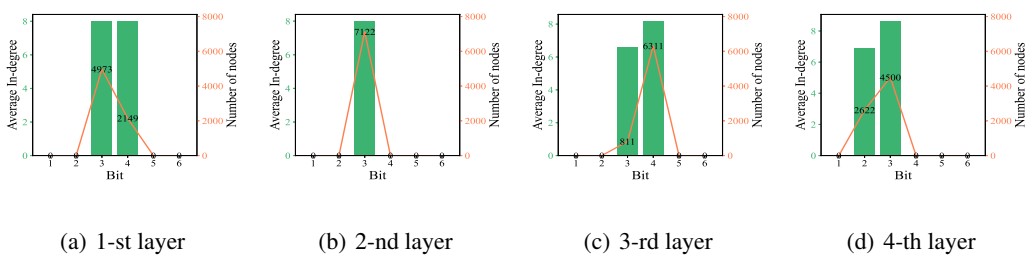

(a) 1-st layer    (b) 2-nd layer    (c) 3-rd layer    (d) 4-th layer

Figure 14: The relationship between bit and average in-degrees of nodes using the corresponding bitwidth to quantize on different layers of GCN trained on MNIST.

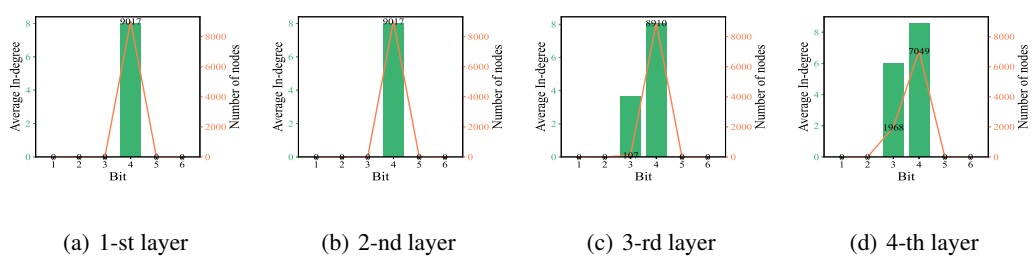

(a) 1-st layer    (b) 2-nd layer    (c) 3-rd layer    (d) 4-th layer

Figure 15: The relationship between bit and average in-degrees of nodes using the corresponding bitwidth to quantize on different layers of GIN trained on MNIST.

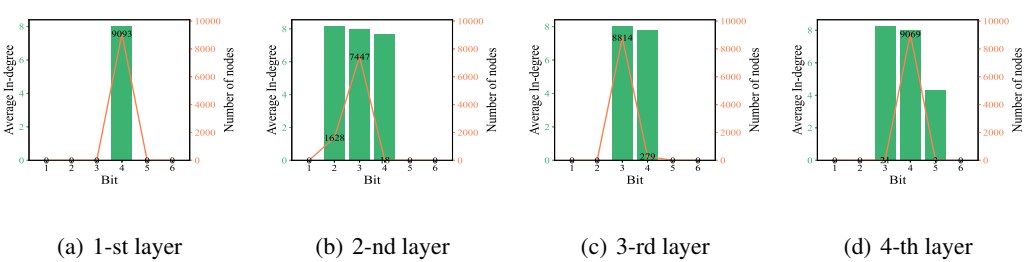

(a) 1-st layer    (b) 2-nd layer    (c) 3-rd layer    (d) 4-th layer

Figure 16: The relationship between bit and average in-degrees of nodes using the corresponding bitwidth to quantize on different layers of GAT trained on MNIST.

Table 13: The impact of the depth of GNNs on quantization performance.

| Layers | | 3 | | 4 | | 5 | |
|---|---|---|---|---|---|---|---|
| Task | | Accu(%) | Avarage Bits | Accu(%) | Avarage Bits | Accu(%) | Avarage Bits |
| GCN-Cora | FP32 | 80.5±0.6 | 32 | 79.3±0.1 | 32 | 75.8±3.2 | 32 |
| | Ours | 80.2±0.6 | 2.94 | 78.2±0.9 | 3.54 | 75.0±1.2 | 3.61 |
| GIN-Cora | FP32 | 49.4±15.8 | 32 | 37.1±13.1 | 32 | —— | —— |
| | Ours | 54.5±12.6 | 3.3 | 36.4±11.1 | 3.1 | —— | —— |

Table 14: The comparison between the model with and without skip connection on GCN-Cora task.

| Layers | GCN-Cora | Without skip connection | | With skip connection | |
|---|---|---|---|---|---|
| | | FP32 | Ours | FP32 | Ours |
| 3 | Accu(%) | 80.5±0.6 | 80.2±0.6 | 82.5±0.5 | 82.2±0.7 |
| | Bits | 32 | 2.94 | 32 | 2.37 |
| 4 | Accu(%) | 79.3±0.1 | 78.2±0.9 | 81.9±0.7 | 81.5±0.3 |
| | Bits | 32 | 3.54 | 32 | 2.63 |
| 5 | Accu(%) | 75.8±3.2 | 75.0±1.2 | 81.1±1.1 | 80.6±0.6 |
| | Bits | 32 | 3.61 | 32 | 2.72 |
| 6 | Accu(%) | 73.8±1.6 | 73.1±1.9 | 80.1±0.8 | 80.4±0.7 |
| | Bits | 32 | 4.62 | 32 | 2.98 |

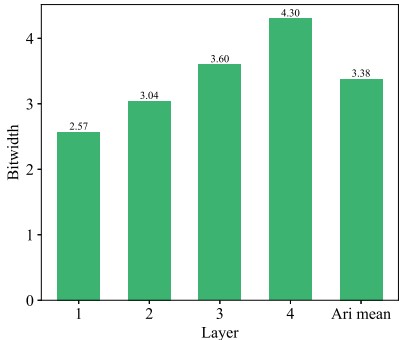

Figure 17: The average bitwidth for 2nd-5th layer in five layers GCN.

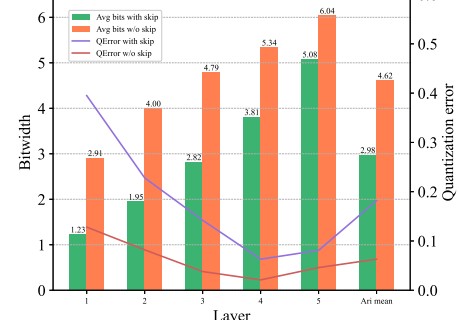

Figure 18: The average bitwidth and quantization error for 2nd-6th layer in six layers GCN.

### A.7.3   MORE ABLATION STUDY

**The impact of the depth of GNNs on quantization performance:** We explore how a different number of GNN layers impacts the quantization performance of GCN-Cora and GIN-CiteSeer. We explore the quantization performance on 3,4,5,6 layers GCN model and 3,4 layers GIN model (the GCN and GIN used in Table 1 are 2 layers). We did not explore the deeper GNN models because

Table 15: The comparison results on other aggregation functions.

| | Baseline(FP32) | Ours | Bit | Compression Ratio |
|---|---|---|---|---|
| GIN_sum | 77.6±1.1% | 77.8±1.6% | 2.37 | 13.5x |
| GIN_mean | 78.8±0.1% | 78.5±0.6% | 2.37 | 13.5x |
| GIN_max | 78.6±1.6% | 78.6±0.5% | 1.97 | 16.2x |

Figure 19: The average aggregated nodes features in different in-degree groups for models with different aggregation functions.

the accuracy of the model decreases drastically as the number of model layers increases due to the over-smooth phenomenon in GNNs. As shown in Table 13, our method can also maintain the performance with a high compression ratio for the model with different layers compared with the FP32 model.

In addition, we observe that the learned quantization bitwidth increases with the number of layers. We analysis the average bitwidth used by 2nd to 5th layer for the five layers GCN model in Figure 17. Our method learns a higher bitwidth for the deeper layer. Due to the over-smooth phenomenon that exists in the deep layer, the embedding features of different nodes are similar in the deep layer. Therefore, we consider the deeper layer may need a higher quantization bitwidth to distinguish the embedding features of different nodes.

**The impact of skip connection on quantization performance:** The first column denoted by 'Without skip connection' and the second column denoted by 'With skip connection' of 18 present the comparison results for different layers GCN on Cora datasets without skip connection and with skip connection, respectively. For the model with skip connection, our method is also effective. Our method learns a higher bitwidth for the deeper layer. Due to the over-smooth phenomenon that exists in the deep layer, we consider that the deeper layer may need a higher quantization bitwidth to distinguish the embedding features of different nodes. and the higher learned quantization bitwidth for deeper layers also alleviate quantization error. And compared to the quantized model with a skip connection, the learned quantization bitwidths are higher for the quantized model without skip connection. Figure 18 presents that the quantization errors of the model with skip connection are always higher than the model without skip connection in every layer which means that the model without skip connection is more sensitive to the quantization error. Therefore, a higher quantization bitwidth is necessary for the model without skip connection to maintain the performance. We will add these analyses to the appendix in the revision.

**Scalability for models that use other aggregation functions:** To demonstrate that our method is also helpful to the GNNs using other aggregation functions rather than the sum function, we replace the aggregation function of the GIN model, which is based on the MPNN framework with mean and max functions, and we conduct the comparison experiment on the Cora dataset. As shown in Table

Table 16: The comparison reults with the binary quantization method on Cora and CiteSeer datasets.

|  |  | Accuracy | Average bits | Compression ratio |
|---|---|---|---|---|
| Cora | GCN(FP32) | 81.5±0.7% | 32 | 1x |
|  | Bi-GCN | 81.2±0.8% | 1 | 32x |
|  | GCN(ours) | **81.4±0.7**% | 1.61 | 19.9x |
|  | GIN(FP32) | 77.6±1.1% | 32 | 1x |
|  | Bi-GIN | 33.7±6.6% | 1 | 32x |
|  | GIN(ours) | **77.4±0.8**% | 1.92 | 16.7x |
|  | GAT(FP32) | 83.1±0.4% | 32 | 1x |
|  | Bi-GAT | 31.9±0% | 1 | 32x |
|  | GAT(ours) | **82.6±0.5**% | 2.03 | 15.8x |
| CiteSeer | GCN(FP32) | 71.1±0.7% | 32 | 1x |
|  | Bi-GCN | 70.7±2.4% | 1 | 32x |
|  | GCN(ours) | 70.7±0.7% | 1.98 | 16.2x |
|  | GIN(FP32) | 66.1±0.9% | 32 | 1x |
|  | Bi-GIN | 29.1±1.7% | 1 | 32x |
|  | GIN(ours) | **65.6±1.5**% | 2.39 | 13.4x |
|  | GAT(FP32) | 72.5±0.7% | 32 | 1x |
|  | Bi-GAT | 20.6±2.6% | 1 | 32x |
|  | GAT(ours) | **71.0±0.7**% | 2.15 | 14.9x |

19, the accuracy degradation is negligible and the compression ratio is high , indicating that our quantization scheme also applies to the GNNs with mean or max aggregation function. We analyze the average features for different aggregation functions in different in-degrees group in Figure 19. The average features of the sum and max functions are highly dependent on in-degrees. The other insight is that the variance of the features is also highly dependent on in-degrees.

The analysis demonstrates the generality of our approach, which can capture differences between nodes introduced by topology information of graphs and compress the model size as much as possible while maintaining the performance.

### A.7.4 COMPARISON WITH BINARY QUANTIZATION METHOD

In this section, we show the advantages of our method over the binary quantization method for GNNs. We select the binary quantization method in Wang et al. (2021b) as our baseline. We just ran the experiments on the node-level because the binary quantization method only supports node-level tasks, which is one of the drawbacks of the binary quantization method in GNNs. We quantize the same part as Wang et al. (2021b) does for a fair comparison.

The comparison results are shown in Table 16. The binary quantization method performs well on GCN, where the aggregation and update phases are simple. However, on both models, GAT and GIN, the accuracy drops significantly compared with the FP32 baseline, which makes the deployment unrealistic. However, our method is immune to this problem, although it has to use a higher average bit for node features which we believe is necessary for GAT and GIN. In summary, our method outperforms the binary quantization method in two ways:

**1.** Our method can quantize more complex GNN models and ensure the accuracy degradation is negligible compared with the FP32 baseline while achieving a high compression ratio of 13.4x-19.9x.

**2.** Our method can be applied to graph-level tasks. However, the binary quantization method can not handle them.

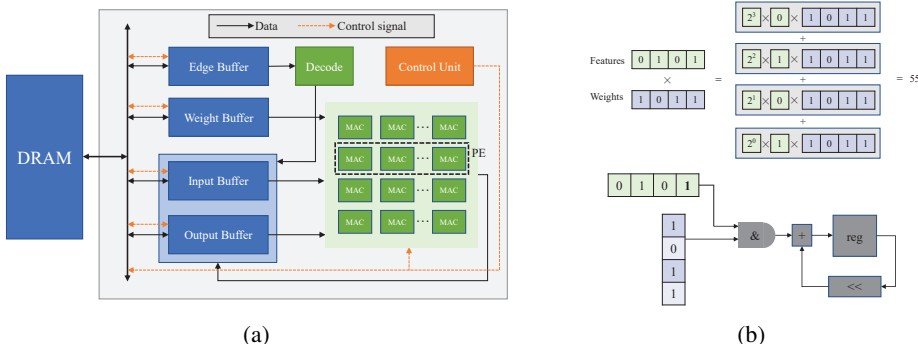

(a)                                                                    (b)

Figure 20: (a) The overview of our accelerator architecture. (b) An example of the bit-serial calculation and the architecture of the MAC.

### A.7.5   ACCELERATOR ARCHITECTURE

In this section, we introduce the architecture of our hardware accelerator designed for GNN inference. As presented in Section 3.1, we quantize each node feature to an appropriate precision and fix the weights to 4bits. To support mixed-precision computation, we adopt bit-serial multipliers at the core. Specifically, we follow the methodology in Judd et al. (2016) to only serialize the node features. This way, it takes $m$ cycles to complete the multiplication between an $m$-bit node feature with a 4bit weight, as shown in Figure 20(b). The product involving $2^n$ is implemented by left-shift, i.e., for $2^n \times a$, we can shift $a$ left by $n$ bits to implement the product. To increase the computational throughput, we use $256 \times 16$ MACs which can process 256 16-dimensional features in parallel. As shown in Figure 20(a), the compute unit is composed of 256 Processing Engines (PEs), each containing a row of 16 MACs. The architecture of the MAC is shown in Figure 20(b).

The on-chip memory consists of an Edge Buffer, which stores the adjacency matrix of graphs, a Weight Buffer, which stores the weight of the GNNs, an Input Buffer, and an Output Buffer to store the input features and the output result, and the register of each MAC to store the partial sum. To reduce data movement in the memory hierarchy, the input buffer and output buffer work in a swapped fashion, as the output of the current layer is the input to the next layer. We set the memory size of Input Buffer, Output Buffer, Edge Buffer, and the Weight Buffer to 2MB, 2MB, 256KB, and 256KB, respectively. The overview of our architecture is shown in Figure 20(a).

To calculate $B^l = X^l W^l$, 256 consecutive rows in $X^l$ and a column of $W^l$ are mapped onto the MAC array to compute 256 inner products in each phase. To achieve this, a column of $W^l$ is broadcast and shared among PEs. The results of the inner products are written to the output buffer, which can be reused to reduce the off-chip DRAM access. The calculation of $X^{l+1} = AB^l$ is also in a inner-product manner. In this scenario, $A$ is a sparse matrix. We therefore represent $A$ in the Compressed Sparse Row (CSR) format, where full zero rows or elements of $A$ are eliminated. During inference, consecutive compressed rows of $A$ and a column of $B^l$ are mapped onto the MAC array in each phase. We also sort the nodes in descending order according to their in-degrees, and the nodes with similar in-degrees are processed in parallel simultaneously to alleviate the load imbalance problem when performing the aggregation operations.

### A.7.6   ENERGY EFFICIENCY ANALYSIS

Our method can save energy cost significantly from the following two aspects:

1. By compressing the model size as much as possible, e.g., 18.6x compression ratio on GCN-Cora as shown in Table 1, our method can significantly reduce the memory footprints. Figure 21 presents the energy table for the 45nm technology node. It shows that memory access consumes further more energy than arithmetic operations. Therefore, the memory footprints domains the energy cost, and then compressing the model can save much energy cost.

| Operation | Energy[pJ] | Relative Cost |
|---|---|---|
| 8 bit int ADD | 0.03 | 1 |
| 8 bit int MULT | 0.2 | 15 |
| 32 bit float ADD | 0.9 | 30 |
| 32 bit float MULT | 3.7 | 123 |
| 32 bit 32KB SRAM | 5 | 167 |
| 32 bit DRAM | 640 | 21333 |

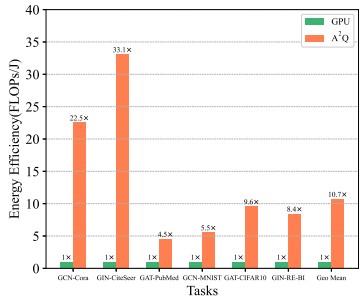

Figure 21: The energy table for 45nm technology node(Han et al., 2016; Sze et al., 2020).

Figure 22: The energy efficiency compared with 2080Ti GPU on various tasks.

2. Through our quantization method and the accelerator, the model can perform inference using the fixed-point operations instead of float-point operations, which are much more energy-consuming than fixed-point operations. As shown in Figure 21, the 32bit float MULT consumes 18.5x energy compared to the 8bit int MULT. Therefore, our method's energy consumption is much lower than the FP32 model.

To illustrate the advantage of our approach in terms of energy efficiency, we compare our accelerator with the 2080Ti GPU on various tasks. To estimate the energy efficiency of GPU, we use the **nvidia-smi** to obtain the power of GPU when performing the inference and measure the inference time by **time** function provided by Python. Then we can get the energy cost of GPU. We also model the energy cost of our method on the accelerator. We use High Bandwidth Memory (HBM) as our off-chip storage. Then we count the number of integer operations, and floating point operations, and the number of accesses to SRAM and HBM when performing the inference process of the quantized models on our accelerator. Based on the data in Table 21, we estimate the energy consumed by fixed-point operations and floating-point operations. The static and dynamic power of SRAM is estimated using CACTI 7.0(Balasubramonian et al., 2017). The energy of HBM 1.0 is estimated with 7 pJ/bit as in (O'Connor, 2014). Figure 22 presents these results, which shows that the the energy efficiency of our method is significantly better than GPU.

