# OpenReview forum: "$\rm A^2Q$: Aggregation-Aware Quantization for Graph Neural Networks"
_ICLR.cc/2023/Conference — ICLR 2023 poster_

### Official Review · Reviewer_wtuK · 2022-10-19

**Confidence:** 5
**Correctness:** 4
**Technical Novelty And Significance:** 3
**Empirical Novelty And Significance:** 3
**Recommendation:** 8

**Clarity, Quality, Novelty And Reproducibility:**

The proposed method is well-explained, and all the implementations detail, including source code, are included.
While some part of the proposed method is based on the previous QAT method, For example, the learning of the step size (section 3.1) was previously proposed by [1]. The entire method, including the non-homogenous bit allocation (different bitwidth for each node)  and prediction of the quantization parameters for unseen nodes, is novel.

[1] https://openreview.net/pdf?id=rkgO66VKDS


**Details Of Ethics Concerns:**

I haven't found any ethical issues.

**Strength And Weaknesses:**

*Strengths*

The paper is well-written, and the method is well-explained.
The evaluation of the proposed method for quantizing graph-level and node-level tasks shows that the proposed method outperforms previous work in terms of compression ratio, computation complexity, and task performance.
The proposed method can works also in a supervised/semi-supervised setting, which allows using the proposed procedure for large-scale real-world graphs. In addition, the authors propose a dedicated GNN accelerator architecture to evaluate the compression ratio and the speedup of the quantization.

*Weaknesses*

Although authors provided evaluations on various graph datasets, all of them are homophilous and transductive. Please elaborate if the proposed method is scaled to non-homophilous or inductive graphs. Supporting experiments would be essential.
The ablation study of how using a different number of GNN layers impacts the quantization performance is not appearing in the current version of the manuscript. I suggest performing this ablation study during the revision stage.

**Summary Of The Paper:**

This paper proposes Quantization-Aware Training (QAT) method, which utilizes the GNN characteristics, e.g.,  topology and nodes attributes.
The proposed method, called  Aggregation-Aware mixed-precision Quantization (A2Q), is automatically learned and assigned
to each node in the graph different bitwidth (number of bits), allowing heterogeneous allocation of different nodes embedding. Furthermore, to mitigate the vanishing gradient problem caused by sparse connections between nodes, the authors propose a local gradient strategy for exploiting the quantization error of the node features as the supervision during the training procedure. In addition, the authors introduce the Nearest Neighbor Strategy for choosing the appropriate quantization settings for unseen graph nodes.
Extensive experiments on eight public node-level and graph-level datasets demonstrate the generality and robustness of our proposed method significantly outperform previous art GNN quantization schemes in terms of cpmp[ression ratio, speedup, and model accuracy.



**Summary Of The Review:**

This paper proposes Quantization-Aware Training (QAT) method, which utilizes the GNN characteristics, e.g.,  topology and nodes attributes.
The paper is well-written, and the method is well-explained.
The evaluation of the proposed method for quantizing graph-level and node-level tasks shows that the proposed method outperforms previous works in terms of compression ratio, computation complexity, and task performance.
While some part of the proposed method is based on the previous QAT method, For example, the learning of the step size (section 3.1) was previously proposed by [1]. However, the entire method is novel, including the non-homogenous bit allocation (different bitwidth for each node)  and prediction of the quantization parameters for unseen nodes.

While overall, I support paper acceptance, there are a few issues that I have pointed out in the weakness section, which I would ask the authors to address during the revision stage.


[1] https://openreview.net/pdf?id=rkgO66VKDS

*****Post-rebuttal review*****
I appreciate the authors for providing a detailed and comprehensive rebuttal and fully responding to my concerns and other reviewers (in my opinion). Hence, I do not doubt that the revised manuscript will contribute to our community, and I would be very happy to see it accepted. My score is also increased to 8.

---

> ### Author Response · Authors · 2022-11-14
> **Response to Reviewer wtuK (Part 1/1)**
>
> Thank you for the positive review and encouraging feedback! We have done experiments
> on more datasets and learning tasks, including the heterogeneous graph, i.e., ogbn-mag and the inductive
> model, i.e., GraphSage. Moreover, we also do the ablation study about the impact of the depth of GNNs on the quantization performance. We have added these to the appendix in our revision.
> We would be happy to add additional clarifications and revisions to the paper to address any additional recommendations from the reviewer.
>
> ### **Q1: Please elaborate if the proposed method is scaled to non-homophilous or inductive graphs.**
>
> Thank you for the constructive advice. **We evaluate our method on inductive
> learning tasks, such as on Flickr and Reddit by using the GraphSage model,
> and also on the heterogeneous
> graph dataset, i.e., ogbn-mag, using the GCN model.** As shown in Table R5-1,
> our method can also compress the model with negligible accuracy degradation.
> And
> on ogbn-mag, our method outperforms the FP32 baseline because the quantization way
> may improve the generalization property of the model to some extent[1].
>
> We have added these comparison results to
> Appendix A.7.3 in the revision.
>
>
>
> **Table R5-1**
> |Tasks|Acc(%)|Bits|
> |:----------------------------:|:----------------------:|:-----------:|
> |GCN-mag|30.8±0.1(FP32)|32|
> ||32.7±0.4(Ours)|2.7|
> |GraphSage-REDDIT|95.2±0.1(FP32)|32|
> ||95.3±0.1(Ours)|3.9|
> |GraphSage-Flickr|50.9±1.0(FP32)|32|
> ||50.0±0.5%(Ours)|3.8|
> |
> >[1] Courbariaux et al., "Binarized neural networks: Training deep neural networks with weights and activations constrained to+ 1 or-1", 2016
>
> ### **Q2: The ablation study of how using a different number of GNN layers impacts the quantization performance is not appearing in the current version of the manuscript.**
>
> We have done this ablation study on GCN-Cora and GIN-Cora tasks. And we have added the experiments to Appendix A.7.3.
> The relevant results are
> listed in Table R5-2. **We explore the quantization performance on 2,3,4,5,6 layers GCN
> model and
> 2,3,4 layers
> GIN model.** We did not explore the deeper GNN models because
> the accuracy of the model decreases drastically as the number of model layers increases due to the over-smooth phenomenon in GNNs.
> **Our method can also maintain the
> performance with a high compression ratio for the model with different layers compared with the FP32 model.**
>
> However, we observe
> that the learned quantization bitwidth increases with the number of layers.
>
> Then we analysis
> the average bitwidth used by 2nd to 5th layer for the five layers GCN model in [Figure R5-1] (https://anonymous.4open.science/r/re_figures-E744/Reviewer5/R5-1.jpg).
> Our method learns a higher
> bitwidth for the deeper layer. Due to the over-smooth phenomenon that exists in the deep layer, the
> embedding features of different nodes are similar in the deep layer. Therefore, we
> consider
> the deeper layer may need a higher quantization
> bitwidth to distinguish the embedding features of different nodes.
>
> Skip connection is a common technology to alleviate the over-smooth phenomenon in GNNs.
> We also explore the impact of skip connection on quantization performance on GCN.
> Table R5-3 presents these results. The insights are two aspects:
> 1. The skip connection can alleviate the over-smooth problem for deep GNN, which is
> also widely
> recognized.
> 2. The skip connection is helpful in compressing the model size.
> With a skip connection, our method
> learns a smaller quantization bitwidth to maintain the performance compared to the model
> without a skip connection (e.g., 2.63 v.s. 3.54 for four layers GCN).
>
> **Table R5-2**
> |Layers||2|layer|3|layer|4|layer|5|layer|6|layer|
> |:---------------|:-----------:|---------------:|:-----------------------|----------------:|:------------------|----------------:|:-----------------------|---------------:|:-----------------------|---------------:|:-----------------------|
> |||Accu(%)|Avarage Bits|Accu(%)|Avarage Bits|Accu(%)|Avarage Bits|Accu(%)|Avarage Bits|Accu(%)|Avarage Bits|
> |GCN-Cora|FP32|81.5±0.7|32|80.5±0.6|32|79.3±0.1|32|75.8±3.2|32|73.8±1.6|32|
> ||Ours|80.9±0.6|1.7|80.2±0.6|2.94|78.2±0.9|3.54|75.0±1.2|3.61|73.1±1.9|4.62|
> |GIN-Cora|FP32|77.6±1.1|32|49.4±15.8|32|37.1±13.1|32|---|---|---|---|
> ||Ours|77.8±1.6|2.37|54.5±12.6|3.3|36.4±11.1|3.1|---|---|---|---|
> |
>
> **TableR5-3**
> |Layers|GCN-Cora|With|skipconnection|
> |:-------------:|:---------------:|:---------------------------------:|:---------------:|
> |||FP32|Ours|
> |3|Acc(%)|82.5±0.5|82.2±0.7|
> ||Bits|32|2.37|
> |4|Acc(%)|81.9±0.7|81.5±0.3|
> ||Bits|32|2.63|
> |5|Acc(%)|81.1±1.1|80.6±0.6|
> ||Bits|32|2.72|
> |6|Acc(%)|80.1±0.8|80.4±0.7|
> ||Bits|32|2.98|
> |

---

> ### Author Response · Authors · 2022-12-03
> **Thanks**
>
> Thanks for your recognition of our work and increasing your score!

---

### Official Review · Reviewer_WF4b · 2022-10-26

**Confidence:** 4
**Correctness:** 4
**Technical Novelty And Significance:** 4
**Empirical Novelty And Significance:** 4
**Recommendation:** 8

**Clarity, Quality, Novelty And Reproducibility:**

The paper is well-written and easy to follow. The proposed methods are well-motivated and novel. Given thorough details, the experiments should be easy to reproduce.

**Strength And Weaknesses:**

Strength:

+ The paper is well-written and easy to follow.

+ The proposed Aggregation-Aware Quantization, Local gradient method, and Nearest Neighbor Strategy techniques are well-motivated and novel.

+ Strong experimental results are shown on a variety of datasets on node-level and graph-level tasks.

Concerns and Questions:

- In Figure 1, the average feature of GAT is invariant to in-degree. Why could GAT benefit from Aggregation-Aware Quantization?

- The average feature depends on the chosen aggregation functions. It would be great to do an analysis and experiments with MPNN[1] or GraphSage [2] with different aggregation functions such as mean, max, and sum. I assume only the average feature of sum aggregation will be highly dependent on in-degree. The question is does GNNs with mean or max aggregations also benefit from Aggregation-Aware Quantization?

- The learnable node-wise quantization has significant computation overhead during the training phase. What is the runtime overhead compared to training a regular GNN?

- I am curious about the degradation of quantization for deep GNNs. The depths of GNNs are not mentioned in the experiment section. The quantization error would be accumulated with more layers. I wonder if this method could be used for deeper GNNs with skip connections.


[1] Gilmer, J., Schoenholz, S.S., Riley, P.F., Vinyals, O. and Dahl, G.E., 2017, July. Neural message passing for quantum chemistry. In International conference on machine learning (pp. 1263-1272). PMLR.

[2] Hamilton, W., Ying, Z. and Leskovec, J., 2017. Inductive representation learning on large graphs. Advances in neural information processing systems, 30.

Reference:

A related work on GNN Quantization: VQ-GNN [3].

[3] VQ-GNN: A Universal Framework to Scale up Graph Neural Networks using Vector Quantization

============================================

 Post rebuttal:
Thanks for the authors' detailed rebuttal and added experiments. I keep my rating unchanged.

**Summary Of The Paper:**

Motivated by the values of node features that are highly related to the graph structure information such as in-degree, the authors propose and learnable quantization method, $A^{2}Q$, to learn the quantization step size and bitwidth for each node in the graph. Local gradient method is used to overcome the vanishing gradient problem for unlabeled nodes in the semi-supervised node level prediction and a Nearest Neighbor Strategy is used to deal with the generalization on unseen
for graph level prediction. Experiments show that $A^{2}Q$ achieves promising results with lower accuracy degradation compared to the previous SOTA on node-level and graph-level tasks.

**Summary Of The Review:**

I would like to recommend a score of 7 for this paper. But there are not 7. So I recommend 8 for now. I will adjust my score depending on the rebuttal.

---

> ### Author Response · Authors · 2022-11-14
> **Response to Reviewer WF4b (Part 2/2)**
>
> ### **Q3: The learnable node-wise quantization has significant computation overhead during the training phase. What is the runtime overhead compared to training a regular GNN?**
>
> [Figure R4-3] (https://anonymous.4open.science/r/re_figures-E744/Reviewer4/R4-3.jpg) compares training time between FP32 models and
> quantized models on various node-level and graph-level tasks. On average, the training time of
> quantized models is 2.04x over the FP32 models.
> Considering that the training time of each epoch varies from 0.003s to 25.41s
> by one 2080Ti GPU, we consider
> this training cost to be acceptable. On the other hand, as shown in [Figure R4-3] (https://anonymous.4open.science/r/re_figures-E744/Reviewer4/R4-3.jpg), the utilization rates of GPU are low, which
> means that there may be room for optimizing the training time.
>
> We also compare the training time on GIN-REDDIT-BINARY(because DQ-INT4 only provides the code on this task)
> with DQ-INT4 using the open source code
> provided by [1]. DQ-INT4 cost 9.27x training time compared with ours.
> >[1] Tailor et al., "Degree-quant: Quantization-aware training for graph neural networks", 2020
>
> ### **Q4: I am curious about the degradation of quantization for deep GNNs. The depths of GNNs are not mentioned in the experiment section. The quantization error would be accumulated with more layers. I wonder if this method could be used for deeper GNNs with skip connections.**
>
> [Figure R4-4] (https://anonymous.4open.science/r/re_figures-E744/Reviewer4/R4-4.jpg) shows the architectures of the models used in our
> evaluations,
> including the layers, the number of hidden units, and whether to use a skip connection.
> The number of heads in the GAT model is 8 in all our experiments.
> We have added this table to Appendix A.6, in which we provide the details about our
> experimental setup.
>
> The question you mentioned about the impact of skip connection on quantization performance
> is an exciting issue. We do a series of experiments on this issue.
>
> The second column, denoted by 'With skip connection' of Table R4-2, lists the comparison
> results for different layers GCN on Cora
> with the skip connection.
> **For the model with different layers, our method can maintain the
> performance with a high compression ratio compared with the FP32 model.**
>
> We observe
> that the learned quantization bitwidth increases with the number of layers.
> To confirm the reasons for this phenomenon,
> we analyze the quantization error and quantization bitiwidth from 2nd to 6th layer
> for the six layers GCN model in [Figure R4-5] (https://anonymous.4open.science/r/re_figures-E744/Reviewer4/R4-5.jpg).
> Our method learns a higher
> bitwidth for the deeper layer. Due to the over-smooth phenomenon that exists in the deep layer,
> we
> consider
> that the deeper layer may need a higher quantization
> bitwidth to distinguish the embedding features of different nodes.
> And the higher learned quantization bitwidth for deeper layers also alleviate
> quantization error.
>
> We also analyze the impact of skip connection on quantization.
> The first column, denoted by 'Without skip connection' of Table R4-2, presents the comparison results for
> different layers GCN on Cora datasets
> without skip connection. Our method is also effective. However, compared to the quantized
> model with a skip connection, the learned quantization bitwidths are higher for the quantized
> model without skip connection. [Figure R4-5] (https://anonymous.4open.science/r/re_figures-E744/Reviewer4/R4-5.jpg) presents that the quantization errors
> of the model with skip connection are always higher than the model without skip connection
> in every layer **which means that the model without skip connection may be more sensitive to
> the quantization error**. Therefore, a higher quantization bitwidth is necessary for the model
> without skip connection to maintain the performance. We have added these analyses to
> Appendix A.7.3 in the revision.
> Hope our responses can address your concerns.
>
> **Table R4-2**
>
> | Layers| GCN-Cora|Without  |skip connection|With |skip connection |
> |:---:|:----:|:---:|:--:|--:|:---:|
> |  ||FP32|Ours|FP32|Ours|
> |3|Acc(%)|80.5±0.6|80.2±0.6|82.5±0.5|82.2±0.7|
> ||Bits|32|2.94|32|2.37|
> |4|Acc(%)|79.3±0.1|78.2±0.9|81.9±0.7|81.5±0.3|
> ||Bits|32|3.54|32|2.63|
> |5|Acc(%)|75.8±3.2|75.0±1.2|81.1±1.1|80.6±0.6|
> ||Bits|32|3.61|32|2.72|
> |6|Acc(%)|73.8±1.6|73.1±1.9|80.1±0.8|80.4±0.7|
> ||Bits|32|4.62|32|2.98 |
> |

---

> ### Author Response · Authors · 2022-11-14
> **Response to Reviewer WF4b (Part 1/2)**
>
> We thank the reviewer for the positive evaluations and the valuable suggestions!
> To address your concerns, we first present the training overhead of our method, and the
> experimental results demonstrate that the overhead is acceptable. We then list the
> architectures of the models used in our experiments. We also explore the impact
> of skip connection on the quantization performance and find some interesting results. Finally,
> our experiments illustrate that the mean and max aggregation can also benefit from our method.
>
> We have revised the paper to fix the issues and to incorporate the suggestions and hope our below response addresses your concerns.
> We would be happy to add additional clarifications and revisions to the paper to
> address any additional recommendations from the reviewer.
>
> In our related work section, we add some works on vector quantization (VQ), such as
> VQ-GNN[1] and EPQuant[2].
>
> >[1] Ding et al., "VQ-GNN: A Universal Framework to Scale up Graph Neural Networks using Vector Quantization", 2021
>
> >[2] Huang et al., "EPQuant: A Graph Neural Network compression approach based on product quantization", 2022
>
> ### **Q1: In Figure 1, the average feature of GAT is invariant to in-degree. Why could GAT benefit from Aggregation-Aware Quantization?**
>
> According to Figure 1 in our paper, the ranges of
> the nodes features in different
> in-degree groups have an upward tendency as the in-degrees increase
> on the GAT model, which means that
> different nodes still have different ranges of features. Therefore, our quantization
> scheme can capture these differences between nodes and learns appropriate
> quantization bitwidth for nodes
> with different feature ranges. And from Figure 4(c), we can also observe that the average
> in-degrees of nodes using different quantization bitwidths are almost invariant, which is
> consistent with the phenomenon that the average feature of GAT is invariant to in-degree,
> i.e., an in-degree group may have features of various ranges, but the variance is large. Figures 8(c) and 11 in our paper also provide the same insight.
>
> ### **Q2: The question is does GNNs with mean or max aggregations also benefit from Aggregation-Aware Quantization?**
>
> Thanks for your enlightening questions!
> We replace the aggregation function of the GIN model, which is based on the MPNN framework,
> with mean and max functions, and conduct the comparison experiment on the Cora dataset.
> Table R4-1 presents the results. The accuracy degradation is negligible and
> the compression ratio is high
> which **indicates that our
> quantization scheme is also applicable to the GNNs with
> mean or max aggregation function.**
>
> We analyze the average features for different aggregation
> functions in different in-degrees
> group in [Figure R4-1] (https://anonymous.4open.science/r/re_figures-E744/Reviewer4/R4-1.jpg). The average features of the sum and max functions are
> highly dependent on
> in-degrees. The other insight is that the variance of the features
> is also highly dependent on in-degrees.
> Therefore, our quantization scheme is also helpful
> to the model with other
> aggregation
> functions by capturing the differences between nodes features.
>
> [Figure R4-2] (https://anonymous.4open.science/r/re_figures-E744/Reviewer4/R4-2.JPG) reveals the
> relationship between quantized bitwidth and average in-degrees of nodes.
> For the GIN model
> with sum and max function, the learned quantization bitwidth
> can present the topology information
> of the graph. Although the learned quantization bitwidth losses the
> topology structure when using the mean function, the model with the mean function can also
> benefit from our quantization scheme due to
> the ranges of features are various.
> We have added this part to our Appendix A.7.3 in the revision
> as a complementary analysis of the main text.
>
> The analysis in Q1 and Q2 demonstrates
> the generality of our approach, which can capture differences between nodes introduced
> by topology information of graphs
> and compress the model size as much as possible while maintaining the performance. We have added these experiments to
> Appendix A.7.3.
>
> **Table R4-1**
>
> |                 |     Baseline(FP32)    |        Ours      |      Bit    |
> |:---------------:|:---------------------:|:----------------:|:-----------:|
> |      GIN_sum    |        77.6±1.1%      |     77.8±1.6%    |     2.37    |
> |     GIN_mean    |        78.8±0.1%      |     78.5±0.6%    |     2.37    |
> |      GIN_max    |        78.6±1.6%      |     78.6±0.5%    |     1.97    |
> |

---

> ### Author Response · Authors · 2022-12-08
> **Thanks**
>
> Thanks for your recognition of our work and the constructive advice which helps us improve the quality of our paper!

---

### Official Review · Reviewer_eih6 · 2022-11-01

**Confidence:** 3
**Correctness:** 3
**Technical Novelty And Significance:** 3
**Empirical Novelty And Significance:** 2
**Recommendation:** 6

**Clarity, Quality, Novelty And Reproducibility:**

The clarity of technical details is not satisfying, should introduce the basic algorithm flow and also see the weakness section above.
The local gradient & NNS techniques are new.
The schemes of learnable quantization parameters and node-wise quantization are not new.

**Strength And Weaknesses:**

==== Strength ====
* Quantizing the node features according to the topology of the graphs can retain higher algorithm performance is reasonable.

* By taking the quantization error of each node as a part of the loss function, the problem that many nodes have no gradient is solved. This is a new method to solve gradient sparsity in graph neural network transductive learning.

* Good to see the code is provided.

==== Weakness and Questions ====

I'm quite confused about Point 1 below. I'd like to see a thorough reply to it.

1. In which scenarios will the proposed method bring meaningful acceleration?

a) Most of the experiments in this paper are transductive learning tasks, which use the information of the whole graph (topology and input features) and labels of some nodes to learn and predict the labels of other nodes. I think for transductive GNN learning tasks, we only need to care about the cost of the training process, since when the training is completed, the original unlabeled nodes "have been" predicted, and there is no inference “after the training”. So, I don't quite understand why we need inference-time-quantization-aware training (QAT) here. The only case that may be useful is that after the training, the topology of the graph doesn’t change but the input features of nodes change, which requires re-prediction. But there are no such experiments. And also, it seems the paper doesn’t verify that there is no need to re-learn the quantization parameters in this case (that is, the quantization parameters only depend on the topology of the graph and are independent of the input features of the nodes of the graph).

b) For the inductive learning scenario, where the Nearest Neighbor Strategy proposed in this paper will be used, the model trained on one graph is used to predict the labels of nodes in other graphs. Because the topology information of other graphs cannot be obtained during training, the basic quantization algorithm proposed in this paper can no longer be used. And the proposed NNS strategy needs to perform floating point inference and determine quantization parameters for each node, to minimize the quantization error. I wonder if the float value of each node has been obtained in this process, the node label of this graph can already be predicted. Why do we need a quantization algorithm?

2. The technical details are not clear enough.

a) The node-wise quantization scheme sets different quantization steps for each node. Then how to add the features of two nodes with different quantization steps during the aggregation? Do we need to operate with floating-point representation for all additions?

b) It's good to see Figure 2 shows how to perform matrix multiplication of features and weights in the node-wise quantization scheme. But the output y will be the input feature x of the next layer. So how can we transform y with different quantization parameters for each element into a format with different quantization parameters for each node? Also, this transformation is carried out online, and its cost should be counted as the cost of inference.

c) Other operations in GNN are not covered, such as the addition of self-feature and aggregation feature, and batch normalization or node degree normalization after aggregation. How are these operations quantized? For example, the normalization operation will have a greater impact on the data distribution after aggregation.

3. Learnable quantization parameters (bit-width, quantization step) are not new for CNN quantization, should refer to more papers on this.

**Summary Of The Paper:**

This paper proposes to learn different quantization parameters for each node during GNN training, so as to reduce the quantization bit width and improve the accuracy. By taking the quantization error of each node as a part of the loss function, it solves the problem that many nodes have no gradient. By making statistics of the maximum value of each node feature on an unseen graph and matching the corresponding parameters, the migration between different graphs is realized without retraining.

**Summary Of The Review:**

My major concerns are listed in the weakness section 1/2, "I'm confused how this method could benefit actual scenario" and "unclear and important details". I hope the authors can explain those points.

---

> ### Author Response · Authors · 2022-11-14
> **Response to Reviewer eih6 (Part 4/4)**
>
> ### **Q2(c): Other operations in GNN are not covered, such as the addition of self-feature and aggregation feature, and batch normalization or node degree normalization after aggregation. How are these operations quantized?**
>
> This answer will also use the notation in Q2(b).
> We have illustrated how to aggregate features in Q2(a). Replacing $A$
> with $\tilde{A}=A+I$ can
> execute the self-feature addition, where $I$ is the identity matrix.
> Then we will illustrate how to quantize the node degree normalization process
> , also using the process of $XW\rightarrow A(XW)$ as an example.
>
> The process of node degree normalization after the aggregation process can be represented
> as
> $X_{l+1} = \sigma(\hat{A}B_l)$, where
> $\hat{A}=D^{-\frac{1}{2}} \tilde{A}D^{-\frac{1}{2}}$ is
> the normalized adjacency matrix,
> and
> $\sigma$ is the non-linear activation function, e.g., ReLU used in our method.
> $D^{-\frac{1}{2}}$ at the right side of $\tilde{A}$ can be fused into
> the $S_{X_l}$ and then calculate $B_{l\_q}$ as Eq. (1) in answer for Q2(b).
> Then the features of the $(l+1)$-th layer $X_{l+1}$ can be obtained as
> $X_{l+1} = \sigma(D^{-\frac{1}{2}} \tilde{A}B_{l\_q})$.
> And there exits
> $X_{l+1} = S_{X_{l+1}}X_{(l+1)\_q}$. Therefore, the $X_{(l+1)\_q}$ can
> be obtained as:
> >$X_{(l+1)\_q} = S_{X_{l+1}}^{-1}X_{l+1}= S_{X_{l+1}}^{-1}\sigma(D^{-\frac{1}{2}}\tilde{A}B_{l\_q})$
>
> Note that the elements in diagonal matrix $S_{X_{l+1}}$ are all positive because
> this matrix is made up of step-size, which is always positive.
> Then we can get
> $X_{(l+1)\_q}=\sigma(S_{X_{l+1}}^{-1}D^{-\frac{1}{2}}\tilde{A}B_{l\_q})$
> , where $S_{X_{l+1}}^{-1}D^{-\frac{1}{2}}$ can be obtained offline and
> $\tilde{A}\in${0,1}$^{N\times N}$. The computation of  $\tilde{A}B_{l\_q}$
> only has addition operations and the $S_{X_{l+1}}^{-1}D^{-\frac{1}{2}}$ can be
> obtained before inference for node-level tasks or introduce only once more element-wise
> multiplication to calculate for the graph-level tasks.
> **The $D^{-\frac{1}{2}}$ at the left side is fused into
> the element-wise multiplication performed by the next layer and
> the $D^{-\frac{1}{2}}$ at the right side
> is fused into the element-wise multiplication performed by the current layer and the
> element-wise multiplication is a necessary stage in the quantized model.** The $AX\rightarrow (AX)W$ calculation way is processed in a similar way. **We have improved the writing about this part in the revision (i.e., Appendix A.3.2).**
>
> When GNNs have Batch Normalization (BN) Layers, the mean and standard-deviation used in BN during the inference
> are obtained from the training process. Therefore, we can fuse the mean and standard-deviation
> with the learned parameters offline. For the i-th column of a matrix, i.e., ${x}$, the mean and standard-deviation
> are $\mu_i$ and $\sigma_i$, and the learned parameters are $\gamma_i$ and $\beta_i$.
> The BN process is:
> >$BN(x)=\frac{x-\mu_i}{\sigma_i}*\gamma_i+\beta_i={x}*\frac{\gamma_i}{\sigma_i}+(\beta_i-\frac{\mu_i} {\sigma_i}*\gamma_i)={x}*y_i+\theta_i$
>
> We can obtain $y_i$ and $\theta_i$ offline.
> Then the calculation process
> is as follows:
> >$X_{l+1} = BN(\sigma(\hat{A}B_{l\_q})) = \sigma(\hat{A}B_{l\_q})Y  + Z$,
>
> where $Y = diag(y_1,y_2,...,y_{F_2})\in \mathbb{R}^{F_2\times F_2}$,
> $Z = (z_1, z_2,...,z_{F_2})\in \mathbb{R}^{N\times F_2}$ and
> $z_i=(\theta_i,\theta_i,...,\theta_i)^T\in \mathbb{R}^N$ among which $y_i$ and
> $\gamma_i$ are the BN parameters for the i-th dimension feature of the features of the nodes.
> And there exists that
> $X_{l+1}=S_{X_{l+1}}X_{l+1\_q}$. Therefore,
> >$X_{l+1\_q} = S_{X_{l+1}}^{-1}X_{l+1}
>  = S_{X_{l+1}}^{-1}(\sigma(\hat{A}B_{l\_q})Y  + Z )
>  = (S_{X_{l+1}}^{-1} \otimes Y) \odot (\sigma(\hat{A}B_{l\_q}))
>  +S_{X_{l+1}}^{-1}Z$.
>
> **Through this process,
> we can fuse the quantization of the next layer into the BN operation of the current layer, which will
> not introduce
> overhead because the BN layer itself requires floating point operations.** Note that the
> float point operations are also element-wise.
>
> Because we consider that fusing quantization with BN may be a common technique in past quantization work [1], we did not perform this derivation in the legacy version.
> **We have added the above derivation process to Appendix A.3.2.**
> Hope our responses can address your concerns.
>
> ### **Q3:Learnable quantization parameters (bit-width, quantization step) are not new for CNN quantization, should refer to more papers on this.**
>
> We have added more references on learnable quantization parameters methods for CNN to the related work section
> in the revision, such as [2,3,4].
> >[1] Li et al., "A System-Level Solution for Low-Power Object Detection", 2019
>
> >[2] Jain et al., "Trained quantization thresholds for accurate and efficient fixed-point inference of deep neural networks", 2020
>
> >[3] Wang et al., "Haq: Hardware-aware automated quantization with mixed precision", 2018
>
> >[4] Esser., "Learned step size quantization", 2019

---

> > ### Comment · Reviewer_eih6 · 2022-11-21
> > **Response**
> >
> > Thanks for the discussion, but I still have remaining concerns, and I think there are some mistakes in the reply.
> >
> > **Q1(a1) & Q1(a2)**
> >
> > Thanks for the added Table R3-1, I think more inductive experiments are necessary for this paper. But, my original concern that "the proposed method cannot accelerate transductive tasks" seems to hold still. In other words, the method can only bring meaningful acceleration for inductive tasks.
> >
> > Actually, for Q1(a2), my original review is trying to propose a scenario where your method can be beneficial for accelerating transductive tasks. The authors might misinterpret the suggestion.
> >
> > **Q1(b)**
> >
> > I agree that dynamic quantization in inference is useful for acceleration in inductive learning scenarios. The NNS to enable quantization on new graphs is to select one from $m$ learned quantization parameters for each node. Then, I have a major concern: instead of $m$ learned parameters, why don’t we generate m predefined $(s, b)$s either uniformly, randomly, or according to the data distribution, and select from them? This is a very important baseline to illustrate the necessity of NNS in inductive scenarios.
> >
> > Also, a minor question: how do you select the 1000 quantization parameters from all the $(s,b)$ groups learned in training?
> >
> > **Q2(a)**
> >
> > I think the aggregation process is $A S_X X_q$ instead of $AX_q S_X$, which cannot be executed using fixed-point calculations. Elements in each row (not column) of $X_q$ belong to one node and share the same scaling factor.
> >
> > **Q2(b)**
> >
> > I was suggesting the authors discuss on the basis of inductive learning scenarios and inference with NNS (since these are the scenario where this method can potentially bring meaningful acceleration). In this case, we can not calculate $S_{X_l} \otimes (S_{W_l}S_{B_l}^{-1})$  offline. We can only calculate $B_l$ by element-wise FP multiplication and then search for the best $S_{B_l}$ to dynamically quantize $B$.
> >
> > **Q2(c)**
> >
> > This particular question in my original review might be somewhat unclear. For example, in GraphSAGE and other networks, if we omit normalization, nonlinear and max-aggregation, we can write the calculation as $Y=AXW_1+XW_2$. This cannot be simply handled by replacing $A$ with $A+I$.

---

> > > ### Author Response · Authors · 2022-11-22
> > > **Response to Reviewer eih6 (Part 2/2)**
> > >
> > > ### **Answer to Q2(a):**
> > >
> > > There were some ambiguities in the original version of our paper about the aggregation phase, which we have modified in the revision during Discussion Stage 1. And **our source code performs exactly in $AX_{q}S_{x}$ way.** As stated in the revision, in the aggregation phase, we quantize $X$ along its feature dimension, i.e., the column direction instead of the row direction, because the nodes features lose the topology information of graphs after the update phase. The GAT and GCN models first update and then aggregate in the same layer. In GIN model, nodes features in the current layer used to aggregate are from the update phase of the previous layer.
> > >
> > > ### **Answer to Q2(b):**
> > >
> > > It is exactly that we cannot compute $(S_{X_l}\otimes (S_{W_l}S_{B_l}^{-1}))$ offline in inductive tasks. The NNS approach introduces more computation of $S_c=S_{W_l}S_{B_l}^{-1}$ and $S_{X_l}\otimes S_c$, which can be calculated offline for the naive quantization process. However, note that the $S_{X_l}$, $S_{W_l}$ and $S_{B_l}^{-1}$ are all diagonal matrices, so the extra float-point multiplications introduced by $S_c=S_{W_l}S_{B_l}^{-1}$ is $\mathcal{O}(F_2)$, and the extra float-point multiplications introduced by $S_{X_l}\otimes S_c$ is $\mathcal{O}(NF_2)$, which is an element-wise float-point multiplication. Therefore, the overhead of float-point multiplication introduced by NNS is negligible as the original answer to [Q1(b)](https://openreview.net/forum?id=7L2mgi0TNEP&noteId=ZdfD1_hsPdT), i.e., $\mathcal{O}(NF_2) \ll\mathcal{O}_I(N^2F_1+NF_1F_2)$. We also analyze the additional float-point multiplication introduced by NNS in [Table R3-2](https://openreview.net/forum?id=7L2mgi0TNEP&noteId=ZdfD1_hsPdT). We can observe that **the overhead introduced by NNS is negligible compared to the overall computation of the model,** i.e., $\mathcal{O}_I(N^2F_1+NF_1F_2)+\mathcal{O}_E(NF_1+NF_2)$.
> > >
> > > On the other hand, by optimizing the pipeline and utilizing the comparator array in our accelerator, the search process also does not incur severe overhead. **Moreover, we have taken the overhead of NNS into account when performing the overall inference process using our accelerator, and our method can also accelerate the inference process compared to DQ-INT4
> > > (results reported in Table 2 of our paper).**
> > >
> > > ### **Answer to Q2(c):**
> > >
> > > In these models, we divide $AXW_1$ and $XW_2$ into two branches and quantize them independently according to our proposed method. It is only necessary to introduce an additional element-wise floating-point addition, i.e., $\mathcal{O}(NF_2)$, in each layer to sum the results of two branches. The overhead is negligible compared to the overall computation complexity, i.e., $\mathcal{O}_I(N^2F_1+NF_1F_2)+\mathcal{O}_E(NF_1+NF_2)$.

---

> > > > ### Comment · Reviewer_eih6 · 2022-12-01
> > > > **Further Response**
> > > >
> > > > Thanks for the clarifications for Q2. They address my concerns about the computation details.
> > > > A remaining issue is that the newly added sentence in the revision, "in which the features lose the topology information of graphs", seems confusing to me. Better modify it to be more rigorous.
> > > >
> > > > Regarding Q1, I still think the reply on "why transductive tasks need inference-time-only acceleration" is not that convincing, as the major computational costs lie in the training process. But I could increase my score if this can be discussed in the limitation section of the paper explicitly.

---

> > > > > ### Author Response · Authors · 2022-12-01
> > > > > **Further response to Reviewer eih6**
> > > > >
> > > > > Thank you for your helpful feedback and appreciation of our work! We would like to express our deepest gratitude for your patience. **As your advice, we will compare the acceleration requirements for inductive, transductive, and supervised tasks, and analyze the limitation of our approach in accelerating the training of transductive tasks where the computational costs are mainly during training in the final version of our paper(Experiments Section).**
> > > > >
> > > > > ### **1.The explaination of "in which the features lose the topology information of graphs"**
> > > > >
> > > > > The aggregation phase and update phase of $l$-th layers are $X_{l+1}=AB_l$ and $B_{l}=X_{l}W_{l}$, respectively. The adjacency matrix $A$ includes the topology information, i.e., degrees of different nodes. Therefore, the result of the aggregation phase, i.e., $X_{l+1}$, is related to the topology of the graphs. **However, for the update phase, i.e., $B_{l}=X_{l}W_{l}$, the result $B_{l}$ is from the transformation of $X_{l}$ using $W_{l}$.** Therefore, the result of the update phase loses topology information of the graphs. We present the relationship between average features and in-degree groups for the result of the update phase on Cora dataset using GCN, GIN, and GAT model in [Figure R3-2] (https://anonymous.4open.science/r/rebuttal_fig2-20F5/R3-2.jpg). **We can observe that the average features obtained from the update phase are independent of the in-degrees.** We will modify this explanation to make it easier to understand in the final version.
> > > > >
> > > > > ### **2.**
> > > > >
> > > > > Our paper focuses on the acceleration of the inference process for various tasks on GNNs. We acknowledge that in transductive tasks, the acceleration of the training process is more urgent than the inference process. Therefore, **as your advice, we will compare the acceleration requirements for inductive, transductive, and supervised tasks, and analyze the limitation of our approach in accelerating the training of transductive tasks where the computational costs are mainly during training in the final version of our paper.**

---

> > > ### Author Response · Authors · 2022-11-22
> > > **Response to Reviewer eih6 (Part 1/2)**
> > >
> > > Thanks for your further discussions and constructive advice on the motivation of training quantization parameters using NNS. I hope the below responses can address your remaining concerns. Also we would be happy to add more comments to address any additional concerns from you.
> > >
> > > ### **Answer to Q1(a1)&Q1(a2)**
> > > * **A1(a1):**
> > >
> > > One of the purposes of the experiments on transductive tasks is to
> > > compare with the previous methods of doing quantization on the inference process for
> > > transductive tasks, e.g., Sgquant[1] and DQ[2].
> > >
> > > **Moreover, we consider that the transductive tasks also need to be accelerated.**
> > > Although the model can obtain the features of unlabeled
> > > nodes after training, the training process uses some techniques to improve the generality,
> > > e.g., dropout and pooling, which result in the features
> > > of unlabeled nodes not being used directly to the downstream tasks after training.
> > > Therefore, it needs to preform inference again using the trained model,
> > > which can be accelerated.
> > >
> > > **In addition, our extensive experiments on node-level and graph-level tasks
> > > illustrate that our approach can accelerate various tasks**, such
> > > as transductive tasks (GCN-Cora and GIN-CiteSeer, etc.), inductive
> > > tasks (GraphSage-Flickr and GraphSage-Reddtit, etc.) and supervised
> > > tasks (GCN-MNIST and GIN-CIFAR10, etc.). **In summary, our approach can
> > > accelerate the forward process of GCN, GIN, GAT, and GraphSage,
> > > which all have a wide range of application scenarios.**
> > >
> > > * **A1(a2):**
> > >
> > > Thanks for your suggestion, but we have yet to find the transductive task that fits this application scenario you mentioned. Tasks on dynamic graphs may be similar to the scenario you mentioned, but it is no longer a pure transductive task. A combination of GNN as an encoder and RNN as a decoder performs the related task for dynamic graphs.
> > >
> > > > [1] Feng et al., "Sgquant: Squeezing the last bit on graph neural networks with specialized quantization", 2020
> > >
> > > > [2] Tailor et al., "Degree-quant: Quantization-aware training for graph neural networks", 2021
> > >
> > > ### **Answer to Q1(b)**
> > >
> > > We first answer the minor question you mentioned.
> > >
> > > Instead of selecting 1000 parameter groups from all (s,b) parameter groups, **we initialize 1000 quantization parameter groups and train these 1000 parameter groups during the training process, and use the NNS to select an appropriate quantization parameter group from the 1000 parameter groups for each node features during the forward process.** The number of parameter groups, i.e., _m_, is an experimental value. In Table 11 of our paper, we explore the impact of different _m_ values on the model performance and select _m_=1000 finally.
> > >
> > > We did comparison experiments with the method of predefined quantization parameters (s,b) without training. We use b=4 as the initialization of bitwidth and do not train the bitwidths. 'Norm' denotes initialization of scale using normal distribution, i.e., $s\in \mathcal{N} (0.01,0.01)$, but no training the quantization parameters and 'Ours-Norm' denotes the initialization using the same normal distribution and training the scale. 'Uniform' and 'Ours-Uniform' are also similar. And the uniform distribution is $s\in \mathcal{U} (0,1)$. The other settings are all the same. **As shown in Table R3-3, we can observe that the accuracy degradation is significant if only predefining the quantization parameters without using NNS to train them.** The method without training scale has less accuracy degradation on REDDIT-BINARY than on CIFAR10 and MNIST, because task on REDDIT-BINARY is a binary classification task and is simpler, while CIFAR10 and MNIST are 10 classification tasks. In addition, we consider that it is not appropriate to initialize quantization parameters based on the data distribution because the distribution of nodes features in different layers is not available in advance. However, our method is able to train the quantization model from scratch, i.e., we do not need any prior knowledge. We will add these analyses to the final version of our paper.
> > >
> > > Table R3-3
> > >
> > > |Task|GCN-CIFAR10|FP32: 55.9±0.4%|GIN-REDDIT-BINARY|FP32: 92.2±2.3%|GAT-MNIST|FP32: 95.6±0.1%|
> > > |--|--|--|--|--|--|--|
> > > |Config|Norm|Ours-Norm|Uniform|Ours-Uniform|Uniform|Ours-Uniform|
> > > |Acc(%)|26.9±0.8|**54.1±0.8**|85.7±1.8|**90.9±2.3**|80.9±2.3|**94.1±0.2**|
> > > |

---

> ### Author Response · Authors · 2022-11-14
> **Response to Reviewer eih6 (Part 3/4)**
>
> ### **Q2(a): ....how to add the features of two nodes with different quantization steps during the aggregation? Do we need to operate with floating-point representation for all additions?**
>
> The calculation of the aggregation process can be represented as matrix
> multiplication, i.e., $AX$. To replace the float point operation with the fixed point operation
> in multiplication, we quantize the $X$ along its feature dimension, i.e.,
> the column direction. Each column of the $X$ has a step-size like the quantization way
> of $W$ in our main text. And $A\in${0,1}$^{N\times N}$ is the adjacency matrix.
> Then the aggregation process can execute as $AX_q S_X$,
> which
> only needs the fixed-point operation and element-wise multiplication($S_X$ is a diagonal matirx). We have modified this corresponding part in the revision.
> How to deal with the node degree normalization, i.e., normalized $A$ matrix,
> lies in Appendix A.3.2 Proof 2, and we will also answer this
> in Q2(c).
>
>
> ### **Q2(b): So how can we transform y with different quantization parameters for each element into a format with different quantization parameters for each node? Also, this transformation is carried out online, and its cost should be counted as the cost of inference.**
>
> After obtaining the $Y$, we will do an element-wise multiplication with the trained
> step-size. Due to the step-sizes being obtained before inference, we can fuse the step-size
> of the current stage with the next stage. Then we use the fused step-size to do an
> element-wise multiplication
> and a truncation operation on bitwidth, which can obtain
> the integer representation of the
> features in the next stage.
> We take the process of $XW\rightarrow A(XW)$ as an illustrative example which
> represents first calculate the $B=XW$ and then calculate $AB$.
> For the $l$-th layer of FP32 models, the first stage is $B_l=X_lW_l$,
> and then calculate the $X_{l+1}=\hat{A}B_l$, where
> $X_l\in \mathbb{R}^{N\times F_1}$, $W_l \in \mathbb{R}^{F_1\times F_2}$ and
> $A\in\mathbb{R}^{N\times N}$.
> The step-size for $B_l$, $X_l$ and $W_l$
> is $S_{B_l}$, $S_{X_l}$ and $S_{W_l}$, respectively.
> And they are all diagonal matrices. The
> more specific form of the step-size matrices can be found in lines 123-138 of our main text.
> The integer representations are calculated as $B_{l}=B_{l\_q}S_{B_l}$,
> $X_{l}=S_{X_l}X_{l\_q}$ and $W_{l}=W_{l\_q}S_{W_l}$.
> Note that for the node-level tasks, we can obtain the
> $S_{B_l}$, $S_{X_l}$ and $S_{W_l}$
> during training. And for the graph-level tasks, we can obtain them through one more
> element-wise multiplication whose overhead is negligible, as the comparison in Table R3-2.
> Then the first stage is
> >$B_l = X_l\cdot W_l = (S_{X_l}\cdot {X_{l\_q}})\cdot (W_{l\_q}\cdot S_{W_l})$,
>
> and
> there exists $B_l = B_{l\_q}S_{B_l}$.
> Therefore, the integers representation for the next stage can be calculated as:
> >$B_{l\_q}  = B_l S_{B_l}^{-1} = (S_{X_l}\cdot {X_{l\_q}})\cdot (W_{l\_q}\cdot S_{W_l})S_{B_l}^{-1}  = (S_{X_l}\cdot {X_{l\_q}})\cdot (W_{l\_q}\cdot (S_{W_l}S_{B_l}^{-1}))=(S_{X_l}\otimes (S_{W_l}S_{B_l}^{-1})) \odot ({X_{l\_q}}\cdot {W_{l\_q}})$ (1),
>
> where the $(S_{X_l}\otimes (S_{W_l}S_{B_l}^{-1}))$ can be calculated offline.
> **Then we obtain the fixed-point representation $B_{l\_q}$ for the next stage and do not introduce
> overhead.** The other calculation way, i.e., $AX\rightarrow (AX)W$, is processed in a similar way.
> There are some contents on how to transform the FP32 features into fixed-point
> representation exists in our Appendix A.3.2
> Proof 2. We have improved the relevant content in the revision.

---

> ### Author Response · Authors · 2022-11-14
> **Response to Reviewer eih6 (Part 2/4)**
>
> ### **Q1(b): I wonder if the float value of each node has been obtained in this process(NNS), the node label of this graph can already be predicted. Why do we need a quantization algorithm?**
>
> We do not quantize
> the FP32 features produced by the last layer,
> which are used for the corresponding application scenarios, such as node classification, edge prediction
> or others.
> Between different layers or different stages
> of the forward process, the float point value
> is exactly obtained.
> **However, we do not use FP32 to perform the forward pass because
> the cost of float-point operations is much more than fixed-point
> operations, whose
> energy cost is less, and latency is lower.** Therefore, we first quantize the features or weights to
> fixed-point values and then do the corresponding calculation using fixed-point values to reduce the
> computation complexity of the models. The overhead introduced by the element-wise float point
> multiplication to transform the value from float-point to fixed-point is negligible.
>
> When performing the Nearest Neighbor
> Strategy (NNS), we fuse this operation with the following operations. The fixed-point results produced by the previous stage
> are used to first multiply the corresponding step-size from
> the previous stage (an element-wise float point multiplication) and
> then execute the NNS process. After getting the step-size,
> these features are quantized immediately (an element-wise float point multiplication)
> and the fixed-point values will be used for the next stage.
> **Therefore, through this fusion way, the FP32 features produced by NNS will not introduce the overhead of storage. And
> this process only needs one more element-wise float-point multiplication.** We present this pipeline in [Figure R3-1] (https://anonymous.4open.science/r/re_figures-E744/Reviewer3/R3-1.jpg).
>
> Then we **illustrate that the overhead introduced by the element-wise float point
> multiplication in NNS is negligible.**
> The computational complexity analysis of the quantized model and FP32 model is as follows:
> For a layer of GNNs, the forward pass can be represented as:
> $X_{l+1}=\hat{A}X_{l}W$, where $\hat{A}\in \mathbb{R}^{N\times N}$,
> $X_{l}\in\mathbb{R}^{N\times F_1}$, and $W_{l}\in \mathbb{R}^{F_1\times F_2}$.
> The
> computational complexity
> of the FP32 models is as follows:
> >$\mathcal{O}(N^2F_1+NF_1F_2)$,
>
> which are all
> the float-point operations. After quantizing
> the model, the float-point
> matrix multiplication can be replaced by integer multiplication and the element-wise
> operation to
> transform the features into an integer representation for the next stage
> (e.g., $AX\rightarrow(AX)W$ or
> $XW\rightarrow A(XW)$),
> which calculates the multiplication between
> integers and float-point numbers according to
> the Eq. 2 in our main text.
> Then
> the computational
> complexity is
> >$C=\mathcal{O}_I(N^2F_1+NF_1F_2)+\mathcal{O}_E(NF_1+NF_2)$,
>
> where $\mathcal{O}_I$ represents the complexity of the integers multiplication, whose cost is much lower
> than the float-point operations,
> and the $\mathcal{O}_E$ represents the complexity of the element-wise
> operations. Note that the $F_1$ is usually 64, 128, 256 or larger
> and $N$ is the number of the nodes
> in a graph or a batch of graphs, so $N$ is always thousands or more. Then
> $N^2F_1\gg NF_1$ and $NF_1F_2\gg NF_2$. **Therefore,
> the overhead introduced by the element-wise multiplication is negligible, and quantization
> can accelerate the inference process and save energy.**
>
> We also
> provide a comparison of the magnitude between INT operation and
> FLOAT operation for the quantized models on various tasks in Table R3-2.
> **From the table, we can observe that the extra float-point operations introduced by NNS is only a tiny fraction of the fixed-point
> operations, e.g., 0.34% in GCN-ZINC.**
>
> On the other hand, through the quantization process, the model size is compressed
> largely, thus the number of memory access will be reduced, which can further save energy and
> accelerate the inference process. Therefore, we quantize the FP32 features and infer using
> the fixed-point representation.
>
> **Table R3-2**
> |Task|GIN-RE-IB|GCN-MNIST|GAT-CIFAR10|GCN-ZINC|
> |-|-|-|-|-|
> |INT(M)|936.96|455.69|1387.98|504.62|
> |FLOAT(M)|7.35|2.06|13.71|1.74|
> |Ratio|0.78%|0.45%|0.98%|0.34%|
> |

---

> ### Author Response · Authors · 2022-11-14
> **Response to Reviewer eih6 (Part 1/4)**
>
> We thank the reviewer for their careful reading of our paper and the constructive questions. Based on the reviewer’s questions, comments, and recommendations, we have made many revisions that may significantly improve the quality of our paper. Addressing the reviewer’s main concern, we do more experiments on the node-level tasks to illustrate
> the significance of our method. We then detail how we quantified the various components in the GNN,
> including the aggregation process, the node degree normalization after aggregation, and the batch
> normalization operations.
>
> Below, we provide point-by-point responses to the reviewer’s feedback and comments and the revisions we made to address them. We would be happy to add additional clarifications and revisions to the paper to address any additional recommendations from the reviewer.
>
> ### **Q1(a1): Most of the experiments in this paper are transductive learning tasks, which use the information of the whole graph (topology and input features) and labels of some nodes to learn and predict the labels of other nodes. I think for transductive GNN learning tasks, we only need to care about the cost of the training process, since when the training is completed, the original unlabeled nodes "have been" predicted, and there is no inference “after the training”. So, I don't quite understand why we need inference-time-quantization-aware training (QAT) here.**
>
> We consider that the transductive tasks you mentioned are the node-level tasks in
> our paper.
> Although our experiments in Table 1 are all transductive learning tasks,
> the purpose of these experiments is to demonstrate that our quantization
> scheme is effective in a computational paradigm, in which the features of nodes
> are generated based on the MPNN framework and ultimately use information from
> elements within the graph, such as edges and nodes rather than the whole graph
> for a range of application scenarios. **Therefore,
> this computational paradigm can also represent the inductive learning tasks at the node level. The four transductive datasets in
> Table 1 in our paper are intended to facilitate comparisons with other quantization methods.**
>
> To illustrate the significance of our approach, we have added some experiments on the
> collab dataset as well as the inductive model, GraphSage, i.e.,
> GCN-collab, GraphSage-REDDIT, and GraphSage-Flickr. These tasks
> all correspond to practical application scenarios, such as recommendation missing citations,
> the prediction
> of user relationships in social networks,
> and the recommendation for related
> products in recommender systems. **Accelerating the inference of GNNs used in these application scenarios
> is meaningful.**
> The collab dataset is a dynamic social network dataset that responds to
> collaborative relationships between different researchers,
> and the goal of this task is to predict whether collaborative
> relationships exist between researchers in different years; Reddit
> is also a social network dataset that it is to predict
> communities of online posts based on user comments;
> Flickr is a task for image classification using the
> description of image attributes.
>
> The results of our experiments are
> presented in Table R3-1, where we can see that our approach still
> works well and even brings some generalization performance
> improvement while significantly compressing the model size. **The experimental
> results on GraphSage demonstrate that our method generalizes well on the inductive model
> for node-level tasks.** We have added these experimental results to Appendix A.7.1.
>
> **Table R3-1**
>
> |             Tasks            |          Acc(%)        |     Bits    |
> |:----------------------------:|:----------------------:|:-----------:|
> |        GraphSage-REDDIT      |      95.2±0.1(FP32)    |      32     |
> |                              |      95.3±0.1(Ours)    |      3.9    |
> |     GraphSage-Flickr    |      50.9±1.0(FP32)    |      32     |
> |                              |     50.0±0.5%(Ours)    |      3.8    |
> |           GCN-Collab         |      44.8±1.1(FP32)    |      32     |
> |                              |      44.9±1.5(Ours)    |      2.5    |
> |
>
> ### **Q1(a2): And also, it seems the paper doesn’t verify that there is no need to re-learn the quantization parameters in this case (that is, the quantization parameters only depend on the topology of the graph and are independent of the input features of the nodes of the graph).**
>
> Table 2 in our main text presents the results on graph-level tasks where the input features of the training and test
> sets are different. These results demonstrate that we do not need to re-learn quantization parameters for different input features.
> In addition, with our proposed Nearest Neighbor Strategy,
> the learned quantization parameters can generalize to other graphs with different topology structures well.

---

> ### Author Response · Authors · 2022-12-01
> **Thanks**
>
> Thanks for your recognition and increasing your score!

---

### Official Review · Reviewer_MZPa · 2022-11-02

**Confidence:** 2
**Correctness:** 3
**Technical Novelty And Significance:** 3
**Empirical Novelty And Significance:** 2
**Recommendation:** 6

**Clarity, Quality, Novelty And Reproducibility:**

Overall the article is decently novel, but the effectiveness would also highly depend on the authors' further clarification.

**Details Of Ethics Concerns:**

No ethics concern

**Strength And Weaknesses:**

Strength:

1. The idea of (in a more deterministic way) utilizing more of the topology information inside of the graph learning is novel, in my opinion.

2. The experiments conducted are extensive.

3. Nice to see the evaluation using tailored hardware!

Weakness:

1. More clarification about the improvement on compression ratio and speed up. In the article, the compression ratio seems to represent the average bits used by the features as compared to FP32. It is better to show (1) more explicit definition of the metric and (2) the reduction in overall memory consumption of the whole inference/training process. For the speedup, it would be great for the authors to more clearly state which part is assumed to run on the accelerator. For instance, would the nearest neighbor part be executed on the accelerator?

2. For NEAREST NEIGHBOR STRATEGY, it seems the FP32 copy of the features is necessary for "the feature with the largest absolute value fi in the node features is first selected, and then we find the nearest qmax"

3. The node-level task accuracy seems too low in Table 1? For instance, https://paperswithcode.com/sota/node-classification-on-cora

4. The authors could provide more illustration on how their methods leverage the graph topology information, as it is a major part of the motivation.

**Summary Of The Paper:**

This work proposes Aggregation-Aware mixed-precision Quantization (A2Q) method to enable an adaptive learning of quantization parameters, which are innovatively linked to the topology of the graph, thus making more use of the graph information. A Local Gradient method is proposed to train the quantization parameters in semi- supervised learning tasks. For unseen nodes,  Nearest Neighbor Strategy to select quantization parameters.

**Summary Of The Review:**

An article which is nicely motivated and target an unique angle of graph learning: how to more pointedly bake in the topology information to help the effectiveness and efficiency. However, a decent amount of clarification is also needed to further recommend this paper.

---

> ### Author Response · Authors · 2022-11-14
> **Response to Reviewer MZPa (Part 2/2)**
>
> ### **Q3: The node-level task accuracy seems too low in Table 1?**
>
> We use the same FP32 baselines as DQ-INT4[1] for a fair comparison.
> These baselines come from papers that propose the corresponding methods.
> To prove that our method is also effective for high-performance models,
> we use the method
> in [2] to obtain higher-performance baselines on GCN-Cora and GCN-PubMed.
> Table R2-2
> lists the comparison results on the higher-performance tasks. Our method can also compress
> the model significantly with negligible accuracy degradation.
>
> **Table R2-2**
>
> |        Tasks      |          Acc(%)        |     Bits    |
> |:-----------------:|:----------------------:|:-----------:|
> |      GCN-Cora     |     87.6±0.1%(FP32)    |      32     |
> |                   |     87.8±0.6%(Ours)    |     2.11    |
> |     GCN-PubMed    |     87.7±0.2%(FP32)    |      32     |
> |                   |     87.2±0.2%(Ours)    |     1.80    |
> |
> >[1] Tailor et al., "Degree-quant: Quantization-aware training for graph neural networks" 2020
>
> >[2] Chen et al., "Fastgcn: fast learning with graph convolutional networks via importance sampling" 2018
>
> ### **Q4: The authors could provide more illustration on how their methods leverage the graph topology information, as it is a major part of the motivation.**
>
> The degree of the nodes is essential topology information of a graph. In the GNNs,
> the higher the
> in-degree is, the larger the features of the nodes tend to be after aggregation. Moreover,
> due to the huge differences between different nodes, the nodes features have
> a wide range of values, as illustrated in Figure 1 in our paper.
> Therefore, it is not
> reasonable to assign the same bitwidth to all nodes,
> which will lead to storage
> waste on nodes with low in-degrees and incur severe quantization errors on nodes with
> high in-degrees.
>
> Our $\rm{A^2Q}$ quantization scheme leverage this property introduced by the
> topology of a graph to learn the appropriate quantization bitwidth for different nodes
> with different in-degrees, which can compress the model size as much as possible and
> maintain the performance simultaneously.
> Our experiments also prove that our method learns
> appropriate bitwidth for the nodes. In Figure 4 of our paper, we provide the relationship
> between
> learned bitwidth and the in-degrees of the nodes. **It shows that our method
> learns more high bitwidths for nodes with high in-degrees
> than those with low in-degrees, which means that our quantization scheme is able to capture
> the differences between node features introduced by the topology of graphs and leverage this property
> to compress the model size as much as possible while maintaining the mdoels
> performance.**
>
> We hope our response can address your concerns.

---

> ### Author Response · Authors · 2022-11-14
> **Response to Reviewer MZPa (Part 1/2)**
>
> Thank you for reviewing our paper and providing helpful feedback on our work.
> We have made many revisions based on the reviewer's questions, comments, and recommendations.
> We update the metric of the compression ratio in the revision and clarify the processing details
> of the FP32 features produced by our NNS method. We also explain how to choose the baseline in our
> experiments
> and further illustrate how our approach leverages the topological information of the graph.
>
> We hope the response below can address your concerns. We would be happy to add additional
> clarifications and revisions to the paper to address any additional recommendations from the reviewer.
>
> ### **Q1: (1) more explicit definition of the metric and (2) the reduction in overall memory consumption of the whole inference/training process. For the speedup, it would be great for the authors to more clearly state which part is assumed to run on the accelerator. For instance, would the nearest neighbor part be executed on the accelerator?**
>
> 1. The metric of compression ratio:
> We analyze the storage required during inference and find that the
> nodes features account for
> 97.64\% - 99.88\% of the overall memory size in FP32 models.
> The bitwidth reported in our paper is the average bitwidth for features in all layers
> of the overall model (line 209). Therefore, we use the ratio between the average bitwidth and 32bit
> (float point) to approximate the compression ratio of the overall memory size.
> For clarity and preciseness, we have updated the metric
> of the compression ratio
> and report the compression ratio of overall memory size in the revision (line 217).
>
>
> 2. The metric of speedup: The **overall inference process** is performed on our accelerator, including
> the nearest neighbor strategy until we obtain the embedding features of the last layer of GNNs.
> Then the embedding features of the last layer are transferred to the host from the accelerator
> to perform the downstream tasks, such as node classification, edge prediction, or graph regression.
> **We have added more clarification about the metric of speedup in the revision (line 212).**
>
> ### **Q2: For NEAREST NEIGHBOR STRATEGY, it seems the FP32 copy of the features is necessary for "the feature with the largest absolute value fi in the node features is first selected, and then we find the nearest qmax"**
>
> Our method does not need to copy FP32 features through our dedicated hardware and the
> optimized pipeline. We present our pipeline in [Figure R2-1] (https://anonymous.4open.science/r/re_figures-E744/Reviewer2/R2-1.jpg). **We fuse the Nearest Neighbor
> Strategy (NNS) with the following operations.** The fixed-point results produced by the previous stage
> are used to first multiply the corresponding step-size from
> the previous stage (an element-wise float point multiplication) and
> then execute the NNS process. After getting the step-size,
> these features are quantized immediately (an element-wise float point multiplication).
> **Therefore, through this fusion way,
> we do not need to store a copy of
> FP32 features.**
>
> In addition, the overhead of the NNS is from one more
> element-wise float point multiplication and the search process.
> We provide a comparison of the number of float-point operations and fixed-point
> operations for different graph-level tasks in Table R2-1, where
> 'INT' denotes the fixed-point operation, 'FLOAT' denotes the float-point operation and
> the 'Ratio' denotes the percentage of the float-point operations in the overall
> process. The extra float-point operations introduced by NNS is only a tiny fraction of the fixed-point
> operations, e.g., 0.34% in GCN-ZINC.
>
> On the other hand, through our optimized pipeline and the comparator array
> used in our accelerator
> the latency introduced by the
> search process of the NNS can be overlapped.
> **Therefore, the overhead introduced by NNS is negligible.**
> We have added this analysis to Appendix A.4 in the revison.
>
> **Table R2-1**
> |     Task         |     GIN-RE-IB    |     GCN-MNIST    |     GAT-CIFAR10    |     GCN-ZINC    |
> |------------------|------------------|------------------|--------------------|-----------------|
> |     INT(M)    |     936.96       |     455.69       |     1387.98        |     504.62      |
> |     FLOAT(M)     |     7.35         |     2.06         |     13.71          |     1.74        |
> |     Ratio        |     0.78%        |     0.45%        |     0.98%          |     0.34%       |
> |

---

> ### Author Response · Authors · 2022-12-04
> **Thanks**
>
> Thank you for your recognition of our work and increasing your score!

---

### Official Review · Reviewer_ZCCp · 2022-11-04

**Confidence:** 3
**Correctness:** 4
**Technical Novelty And Significance:** 3
**Empirical Novelty And Significance:** 3
**Recommendation:** 8

**Clarity, Quality, Novelty And Reproducibility:**

The writing is clear. However, I have concerns regarding the novelty of the approach, as explained above. I have not tried to reproduce the results.

**Strength And Weaknesses:**

Strengths:

1. Given the increasing deployment of GNNs in edge devices, it is important to improve their energy efficiency. This paper tackles this timely problem, and is generally well-motivated.

Weaknesses:

1. My biggest concern is the limited novelty of the paper. It seems that the paper is applying known tricks in mixed-precision quantization to GNNs. The authors add a statement in section 2 "However,  due to the huge difference between GNNs and CNNs, it is difficult to use these methods on GNNs directly". However, I am not convinced with such an argument without concrete evidences. What happens if these techniques are naively applied to GNNs?
2. The paper only reports speedup with their quantization scheme. What about the energy/power consumption which is another important metric given the resource constraints of edge devices?
3. The authors do not quantify their accuracy-bitwidth trade-off with several related works they cite in section 2. Only DQ-int4 is considered.
4. Several related comparisons with quantized GNNs are missing, such as [1-3].

[1] Brennan et al., "Not Half Bad: Exploring Half-Precision in Graph Convolutional Neural Networks", 2020

[2] Huang et al., "VEPQuant: A Graph Neural Network compression approach based on product quantization", 2022

[3] Zhao et al., "LEARNED LOW PRECISION GRAPH NEURAL NETWORKS", 2020

**Summary Of The Paper:**

This paper proposes an aggregation-aware mixed-precision quantization method for GNNs, which quantizes different nodes features with different learnable quantization parameters, including bit-width and step-size. The paper also proposes a Nearest Neighbor Strategy to deal with the generalization on unseen graphs. Empirical evaluations on several node-level and graph-level datasets demonstrate the efficacy of the proposal.

**Summary Of The Review:**

See weaknesses above. I am giving a rating of 3 for now, but I can reconsider my rating based on the authors' rebuttal and discussions with the other reviewers.

*******************************

Review Update: Score changed to 8 after rebuttal.

---

> ### Author Response · Authors · 2022-11-14
> **Response to Reviewer ZCCp (Part 3/3)**
>
> ### **Q3: The authors do not quantify their accuracy-bitwidth trade-off with several related works they cite in section 2. Only DQ-int4 is considered.**
>
> Table R1-4 lists the comparison results
> with the DQ-INT8, which
> uses 8bit to quantize the model in DQ, on GCN-Cora and GIN-CiteSeer tasks using
> different learned quantization bitwidth of our method.
> **By turning down the penalty factor
> on $L_{memory}$, our method performs better than DQ-IN8 using a smaller quantization
> bitwidth (e.g., 3.89 v.s. 8).**
> In our main text, for a better experimental results comparison, we only consider the SOTA method in
> muti-bit quantization, and
> the DQ-INT4 is the SOTA results in 4bit quantization. Therefore, we compare our method
> with the DQ-INT4, and we think it is sufficient to prove the effectiveness of our method.
> On the other hand, **some results reported in Table 1 of our main text have obtained
> comparable and even better performance using a smaller bit compared with DQ-INT8
> (e.g., on GAT-CiteSeer or GAT-Cora tasks).**
> And as the advice, we also
> compare with the works you mentioned in Table R1-5.
> What is more, we also have provided the ablation study about
> the accuracy-bitwidth trade-off of our method in Section 5,
> "The advantage of learning-based mixed-precision quantization" and the results
> in Table R1-4 can also reveal the accuracy-bitwidth trade-off relationship.
>
> **Table R1-4**
>
> |     GCN-Cora    |               |     FP32:81.5±0.7%    |                 |     DQ-INT8:81.7±0.7%    |                 |
> |:---------------:|:-------------:|:---------------------:|:---------------:|:------------------------:|:---------------:|
> |       Ours      |     Accu(%)    |        80.9±0.6       |     81.2±0.4    |          82.0±0.3       |     82.2±0.2    |
> |                 |      Avg bits     |          1.70         |       2.28      |            3.89          |       4.48      |
> |     GIN-Cite    |               |     FP32:66.1±0.9%    |                 |      DQ-INT8:67.5±1.4%    |                 |
> |       Ours      |     Accu(%)    |        65.1±1.7       |     66.4±0.4    |          67.2±0.3        |     68.2±0.8    |
> |                 |      Avg bits     |          2.54         |       2.88      |            3.94          |       4.37      |
> |
>
>
> ### **Q4:Several related comparisons with quantized GNNs are missing**
>
> Thank you for making the part of the related work in our text more complete.
> Table R1-5 presents
> the comparison results on various tasks in [1] and [3]. Our method can perform better
> on all tasks with a smaller quantization bitwidth.
>
>
> We do not compare with the method in
> [2], because we consider that [2] uses a different way to compress the model size, i.e.,
> Product Quantization (PQ), and a fair comparison with this method is not realistic because
> the product quantization method need to search for the vector in a codebook which is  computationally complex. In addition, our method is orthogonal
> to the PQ. Therefore, the two methods can be applied to compress model size simultaneously.
>
> We have added [3] to the related work part to make our paper more complete and we also
> add the comparsion experiments with [1,3] to Appendix A.7.1 in the revision.
>
> **Table R1-5**
>
> |           Task          |         Accu(%)        |         Avg bits       |
> |:-----------------------:|:---------------------:|:------------------:|
> |         GCN-Cora        |       80.9±0.0[1]     |     A-16   W-16    |
> |                         |     **80.9±0.6**(Ours)    |     **A-1.7   W-4**    |
> |       GAT-CiteSeer      |       68.0±0.1[3]     |      A-8   W-8     |
> |                         |     **71.9±0.7**(Ours)    |     **A-1.9   W-4**    |
> |      GraphSage-Cora     |       74.3±0.1[3]     |     A-12   W-10    |
> |                         |     **74.5±0.2**(Ours)    |     **A-2.7   W-4**    |
> |     GraphSage-Flickr    |       49.7±0.3[3]     |      A-8   W-4     |
> |                         |     **50.0±0.5**(Ours)    |     **A-3.8**   W-4    |
> |
>
>
>
> >[1] Brennan et al., "Not Half Bad: Exploring Half-Precision in Graph Convolutional Neural Networks", 2020
>
> >[2] Huang et al., "VEPQuant: A Graph Neural Network compression approach based on product quantization", 2022
>
> >[3] Zhao et al., "LEARNED LOW PRECISION GRAPH NEURAL NETWORKS", 2020

---

> ### Author Response · Authors · 2022-11-14
> **Response to Reviewer ZCCp (Part 2/3)**
>
> ### **Q2: "What about the energy/power consumption which is another important metric given the resource constraints of edge devices?"**
>
> Our method can save energy cost significantly from the following two aspects:
>
> 1. By compressing the
> model size as much as possible, e.g., 18.8x compression ratio on GCN-Cora as
> shown in Table 1 in our paper,
> our method can significantly reduce the memory footprints.
> Table R1-3 presents the energy table for the 45nm technology node[1,2].
> It shows that memory access consumes further more energy than
> arithmetic operation. **Therefore,  compressing the model can save much energy cost.**
>
> 2. Through our quantization method and the accelerator, the model can perform inference
> using the fixed-point operations instead of float-point operations which are much more energy-consuming than fixed-point operations. As shown in Table R1-3,
> **the 32bit float MULT consumes 18.5x energy compared to the 8bit int MULT.**
> Therefore, the energy consumption of our method is much lower compared with the FP32 model.
>
> To illustrate the advantage of our approach in terms of energy efficiency, we compare our
> accelerator with the 2080Ti GPU on various tasks.
>
> To estimate the
> energy efficiency of GPU, we use the `nvidia-smi` to obtain the power of 2080Ti GPU
> when performing the inference and
> measure the inference time by `time` function provided by Python. Then we can get the energy cost of
> GPU.
>
> We also model the energy cost of our method on the accelerator.
> We use High Bandwidth Memory (HBM) as our off-chip storage.
> Then we count the number of fixed-point operations,
> and float-point operations, and the
> number of accesses to SRAM and HBM when performing the inference process of the quantized models
> on our accelerator.
> Based on the data in Table R1-3, we
> estimate the energy consumed by fixed-point
> operations and float-point operations.
> The static and dynamic power of SRAM is estimated using CACTI 7.0[3].
> The energy
> of HBM 1.0 is estimated with 7 pJ/bit as in [4].
> [Figure R1-1] (https://anonymous.4open.science/r/re_figures-E744/Reviewer1/R1-1.jpg)
> presents these results, which
> shows that
> the
> the energy efficiency of our method is significantly better than GPU. **On average, our method achieves 10.7x energy saving compared to
> the 2080Ti GPU.**
> We have added this
> content to Appendix A.7.6 in the revision.
>
> **Table R1-3**
>
> |     Operation              |     Energy[pJ]    |     Relative   Cost    |
> |----|--|---|
> |     8   bit int ADD        |     0.03          |     1                  |
> |     8   bit int MULT       |     0.2           |     15                 |
> |     32   bit float ADD     |     0.9           |     30                 |
> |     32   bit float MULT    |     3.7           |     123                |
> |     32   bit 32KB SRAM     |     5             |     167                |
> |     32   bit DRAM          |     640           |     21333              |
> |
> >[1] Han et al., "EIE: Efficient inference engine on compressed deep neural network", 2016
>
> >[2] Sze et al., "Efficient processing of deep neural networks", 2017
>
> >[3] Balasubramonian et al., "CACTI 7: New tools for interconnect exploration in innovative off-chip memories", 2017
>
> >[4] O'Connor et al., "Highlights of the high-bandwidth memory (hbm) standard", 2014

---

> ### Author Response · Authors · 2022-11-14
> **Response to Reviewer ZCCp (Part 1/3)**
>
> Thank you for your detailed feedback. We want to clarify in detail the main concern
> about the novelty of our method with extensive experiments and analysis. Moreover, we
> compare the energy efficiency with GPU to demonstrate the advantage of the energy efficiency of our method. We also
> make the related work section more complete and do more comparison experiments with more GNN quantization methods based on your recommendations.
>
> We have
> revised the paper to fix the issues and incorporate the suggestions.
> We hope the response below can address your concerns.
> We would be happy to add additional clarifications and revisions to the paper to
> address any additional recommendations from the reviewer.
>
> ### **Q1: "My biggest concern is the limited novelty of the paper. ...... What happens if these techniques are naively applied to GNNs?"**
>
> We do the comparison experiments with the naive method on the GCN-CiteSeer, GIN-Cora and
> GIN-REDDIT-BINARY tasks. The results in Table R1-1 and Table R1-2
> demonstrate that the accuracy degradation is significant
> using the naive method. **There are mainly two challenges when applying the mixed-precision technology
> directly to the GNNs.**
>
> 1. Most real-world graphs often follow the power-law
> distribution[1,2],
> i.e., the nodes with a low degree account for the majority of graph data, which incurs the
> adjacency matrix of the graph often being extremely sparse. Moreover, in the semi-supervised learning
> task where the GNNs are commonly applied, only a tiny
> fraction of nodes have a label (e.g., 0.30\% in the PubMed dataset). The above
> two reasons lead to **sparse gradients** of the loss function with respect to the
> quantization parameters.
> **Therefore, the naive method can not train the quantization parameters
> well, and we propose the Local Gradient method to solve this problem.**
>
> Figure 3 in our paper illustrates the sparse gradient phenomenon
> using
> GCN-Cora task as an example, and we also provide a rigorous
> proof of this phenomenon in Appendix A.3.2.
>
> Table R1-1 presents the comparison results between our method and the naive method
> on Cite and Cora.
> The naive method suffers from a significant accuracy degradation and can not learn the bitwidth well
> (e.g., all nodes use 3bit on GCN-Cite). However,
> our local gradient
> method overcomes these drawbacks.
>
> **Table R1-1**
>
> |       Task      |     Config    |        Accu      |     Avg   bits    |
> |:--:|:---:|:--------:|:---:|
> |     GCN-Cite    |      FP32     |     71.1±0.7%    |         32        |
> |                 |      Naive    |     56.8±6.7%    |          3       |
> |                 |      Ours     |     **70.6±1.1%**    |        **1.87**       |
> |     GIN-Cora    |      FP32     |     66.1±0.9%    |         32        |
> |                 |      Naive    |     42.1±1.4%    |          4        |
> |                 |      Ours     |     **65.1±1.7%**    |        **2.54**       |
> |
> >[1] Xie et al., "Distributed power-law graph computing: Theoretical and empirical analysis", 2014
>
> >[2] Aiello et al., "A random graph model for power law graphs", 2001
>
> 2. When processing the graph-level (e.g., GIN-REDDIT-BINARY) tasks,
> the number of nodes input to the GNNs varies.
> **However, the input to CNNs is always in a fixed size; therefore, the naive method
> can only train a fixed number of quantization parameters, which is not applicable to GNNs.**
> We propose the Nearest Neighbor Strategy (NNS) to make it feasible to
> quantize the graph with a variable number of nodes using a fixed number of quantization parameters.
>
> We compare the naive method with our NNS method in Table R1-2. It shows that the naive
> method can not apply directly to GNNs due to the significant degradation of accuracy.
>
> The second row of Tabel 3 and the "The overhead of Nearest Neighbor Strategy" in Section 5
> in
> our paper also present
> these comparisons.
> We hope our response can address your concerns.
>
> **Table R1-2**
>
> |            Task          |     Config    |        Accu      |     Avg   bits    |
> |:-----:|:---:|:--------:|:-------:|
> |     GIN-REDDIT-BINARY    |      FP32     |     92.2±2.3%    |         32        |
> |                           |      Naive    |     71.5±5.4%    |          4        |
> |                           |      Ours    |     **90.8±1.8%**    |         **3.5**       |
> |

---

> ### Comment · Reviewer_ZCCp · 2022-12-04
> **Response**
>
> The authors have addressed all my concerns, in particular the one about novelty. I am increasing my score from 3 to 8. I recommend the authors highlight the novelty and include the energy efficiency results in the final version.

---

> > ### Author Response · Authors · 2022-12-04
> > **Response and Thanks**
> >
> > Thank you for recognizing our work and increasing your score! Your
> > advice and comments have indeed enabled us to improve the quality of our
> > paper. We have added the energy efficiency analysis to the revision
> > and will highlight the novelty of our approach further in the final version.

---

### Author Response · Authors · 2022-11-14
**General response to PC, AC, and all reviewers**

Thank you reviewers for your helpful feedbacks and the constructive advice. Based on the reviewers' questions, comments, and recommendations, we have made many revisions that may significantly improve the quality of the paper.

We have submitted our revision and highlighted the changes.

---

### Author Response · Authors · 2022-11-17
**Looking forward to further discussions!**

Dear Reviewers,

We were wondering if our response and revision have addressed all your concerns. As we noted in our previous responses, we conducted additional experiments and attempted to address all of your concerns. In the remaining two days of the discussion Stage 1, we would appreciate it if you could let us know whether you have any other questions, so that we can still have time to respond and address them. We are looking forward to discussions that can further improve our current manuscript. Thanks!

Best regards,

The Authors

---

### Author Response · Authors · 2022-11-20
**Summary of newly conducted experiments and revisions.**

* [Reviewer ZCCp:](https://openreview.net/forum?id=7L2mgi0TNEP&noteId=fH3c5i43Sw) Following your comments, we compare the energy efficiency with 2080Ti GPU and have added these results to Appendix A.7.6. We also compare with more quantization methods you mentioned (Appendix A.7.1) and modify our related work section (Section 2). Finally, we present the accuracy-bitiwidth trade-off results compared with DQ-INT8 and the trade-off results of our method located in Section 5.

* [Reviewer MZPa:](https://openreview.net/forum?id=7L2mgi0TNEP&noteId=lXMHHeXgW44) Following your comments, we modified the metric of compression ratio and clarified which part is assumed to run on our accelerator in Section 4.1. Moreover, we have added more detailed information about the overhead analysis on NNS to Appendix A.4 to illustrate the NNS process. We also demonstrate how our methods leverage the graph topology information in our comments.

* [Reviewer eih6:](https://openreview.net/forum?id=7L2mgi0TNEP&noteId=cXUFUNcYVo) Following your comments, we have added more node-level experiments on inductive learning tasks to Appendix A.7.1, which can explain the significance of our methods. We have modified the content about how to aggregate in Section 2. In addition, to illustrate the quantization process of various components of GNNs, e.g., degree normalization and batch normalization, more clearly, we also modified the content in Appendix A.3.2. We also modified our related works in Section 2.

* [Reviewer WF4b:](https://openreview.net/forum?id=7L2mgi0TNEP&noteId=emwGxir_wbH) Following your comments, we conducted the experiments about whether GNNs using different aggregation functions can also benefit from our quantization method and explore the impact of different layers on the quantization performance. In addition, we also analyze the impact of skip-connection on our quantization method. We have added all the above experiments to Appendix A.7.3. We also modified our related works in Section 2.

* [Reviewer wtuK:](https://openreview.net/forum?id=7L2mgi0TNEP&noteId=8DtP1jJC42) Following your comments, we conduct more experiments on non-homophilous graph, e.g., ogbn-mag, and the inductive learning tasks, e.g., GraphSage model on Reddit and Flickr. Constrained by the length of the main text, we have added these experiments to Appendix A.7.1. In addition, we conduct the ablation study of how using a different number of GNN layers impacts the quantization performance. We have added these results to Appendix A.7.3.

---

### Decision · Program_Chairs · 2023-01-20

**Decision:**

Accept: poster

**Justification For Why Not Higher Score:**

Reviewers also expressed concerns about the limited novelty of this paper, although the reviewers convinced the committee that there is vanishing gradient problem cause by sparse connections between nodes which is addressed by the proposed local gradient method. However, the justification of this local method is not clearly illustrated.

**Justification For Why Not Lower Score:**

The reviewers liked the idea of utilizing more of the topology information inside of the graph learning. And the problem considered is timely.

**Metareview: Summary, Strengths And Weaknesses:**

The paper considers an important problem in the graph community, scalability during inference/deployment time. The authors propose an Aggregation-Aware mixed-precision Quantization, which quantizes different nodes features with different learnable quantization parameters, including bit-width and step-size. To mitigate the vanishing gradient problem caused by sparse connections between nodes, the authors propose a Local Gradient method to serve the quantization error of the node features as the supervision during training. The paper also proposes a Nearest Neighbor Strategy to deal with the generalization on unseen graphs.

The reviewers liked the idea of utilizing more of the topology information inside of the graph learning. However, reviewers also expressed concerns about the limited novelty of this paper, although the reviewers convinced the committee that there is vanishing gradient problem cause by sparse connections between nodes which is addressed by the proposed local gradient method. However, the justification of this local method is not clearly illustrated.



**Note From Pc:**

if the above contains the word "oral" or "spotlight" please see: "oral" presentation means -> notable-top-5% and "spotlight" means -> notable-top-25%. As stated in our emails, we are disassociating presentation type from AC recommendations